# Learning from Snapshots of Discrete and Continuous Data Streams

**Pramith Devulapalli**
Department of Computer Science
Purdue University
pdevulap@purdue.edu

**Steve Hanneke**
Department of Computer Science
Purdue University
steve.hanneke@gmail.com

## Abstract

Imagine a smart camera trap selectively clicking pictures to understand animal movement patterns within a particular habitat. These "snapshots", or pieces of data captured from a data stream at adaptively chosen times, provide a glimpse of different animal movements unfolding through time. Learning a continuous-time process through snapshots, such as smart camera traps, is a central theme governing a wide array of online learning situations. In this paper, we adopt a learning-theoretic perspective in understanding the fundamental nature of learning different classes of functions from both discrete data streams and continuous data streams. In our first framework, the *update-and-deploy* setting, a learning algorithm discretely queries from a process to update a predictor designed to make predictions given as input the data stream. We construct a uniform sampling algorithm that can learn with bounded error any concept class with finite Littlestone dimension. Our second framework, known as the *blind-prediction* setting, consists of a learning algorithm generating predictions independently of observing the process, only engaging with the process when it chooses to make queries. Interestingly, we show a stark contrast in learnability where non-trivial concept classes are unlearnable. However, we show that adaptive learning algorithms are necessary to learn sets of time-dependent and data-dependent functions, called pattern classes, in either framework. Finally, we develop a theory of pattern classes under discrete data streams for the blind-prediction setting.

## 1 Introduction

### 1.1 Two Motivating Examples

Pretend you're a farmer by day and businessperson by night. As a farmer, you oversee a 10,000 acre plot of land equipped with a smart irrigation system. To feed data to your irrigation system, you rely on hyperspectral imaging taken from a satellite to gauge soil moisture conditions. Ideally, you would like to constantly feed your irrigation system with hyperspectral data; however, the steep financial cost of processing hyperspectral data prevents you from doing so. As a result, you need to devise a strategy to sparingly use satellite data; at all other times, you rely on the smart irrigation system to accurately extrapolate the soil moisture conditions as time passes by.

At night, you become a businessperson. You employ a translator on your work laptop during your virtual meetings to automatically convert your voice into the preferred language of your client. This translator is fine-tuned by a speech-to-text translation system that takes in voice data and updates the translator's model on the correct language translation. But, there's a caveat. Each request costs money. And each transmission dominates a sizable portion of the available Internet bandwidth. Your task is to come up with the optimal strategy of balancing requests to the cloud versus trusting the fidelity of the translator.

38th Conference on Neural Information Processing Systems (NeurIPS 2024).

## 1.2 A New Learning Paradigm

While these settings may seem rather creative in nature, both of these scenarios represent plausible real-world instances of learning from continuous data streams with temporal dependencies. What type of learning-theoretic framework should one construct when framing the question of online learning under continuous data streams? How can we best capture the notion of temporal dependencies and patterns that naturally arise when analyzing such data sources? While these questions are highly pertinent, the answers aren't clear due to a vast majority of the learning theory literature focusing on online learnability from discrete data streams modeled as round-by-round processes. In the two examples showcased at the beginning, it's clear that establishing a theoretical understanding of these settings can be an important step in tackling online learnability under continuous data streams.

In our paper, we present a streamlined approach in tackling these rather fundamental challenges by first establishing two closely related, but separate, frameworks.

**Blind-Prediction Setting**  The first framework is called *blind-prediction* which is highlighted by the smart irrigation system using satellite imagery data. The irrigation system receives feedback only when hyperspectral data is requested; at all other times, the system must predict on its own with no input from the environment. This framework is designed such that a learning algorithm must make a prediction based only on the current timestamp and previous queries. The learner's predictions are independent of the current values generated by the data stream hence the name *blind-prediction*.

**Update-and-Deploy Setting**  The second framework, called *update-and-deploy*, is highlighted by the speech-to-text translation system. The speech-to-text translation system, a learning algorithm, and the translator, called the predictor, are considered as two separate entities where the algorithm retrieves snapshots of the data stream to update the predictor. We describe this behavior as a learning algorithm activating different modes at different times. A learning algorithm performs *updates* to a predictor when it queries and *deploys* the predictor to make predictions as the process rolls by.

**Pattern Classes**  A significant portion of this work is dedicated to studying these frameworks under *pattern classes*, sets of sequences encoding data-dependent and time-dependent characteristics. First introduced by Moran et al. [1], these classes consist of a set of patterns; each pattern is a sequence of instance-label pairs marked with the appropriate timestamp. For example, if we let $\mathcal{X}$ and $\mathcal{Y}$ represent the instance space and label space respectively, then $Z^\infty = (\mathcal{X} \times \mathcal{Y})^\infty$ represents the set of all countably infinite patterns. A discrete pattern class $\mathcal{P}$ is defined as $\mathcal{P} \subseteq Z^\infty$ where any $P \in \mathcal{P}$ is understood as $P = (Z_t)_{t=1}^\infty = (X_t, Y_t)_{t=1}^\infty$.

Pattern classes can also be viewed as natural generalizations of concept classes. Given a concept class $H$ consisting of classifiers mapping instances from $\mathcal{X}$ to labels in $\mathcal{Y}$, we can derive a pattern class that encapsulates all sequences that could be realized by any single $h \in H$. Formally speaking, the induced pattern class $\mathcal{P}(H)$ is defined as $\mathcal{P}(H) = \{(Z_t)_{t=1}^\infty \in Z^\infty : \exists h \in H, \forall t \in \mathbb{N}, h(X_t) = Y_t\}$

Now that the stage has been developed for pattern classes, we turn to a set of questions that naturally arise under continuous data streams. What pattern classes are online learnable? Is there a natural dimension that characterizes online learnability of pattern classes under the different querying-based models? How does learning pattern classes and concept classes differ under continuous data streams? We tackle these important questions in our paper using our learning frameworks.

## 1.3 Our Contributions

We detail the primary contributions of this work below.

1. **Non-Adaptive Learners in the Update-and-Deploy Setting.** First, we extend the current theory on concept classes to include online learning under continuous data streams for the update-and-deploy setting. A non-adaptive learner is a learning algorithm that queries independent of the process itself. For the *update-and-deploy* setting, we show that the non-adaptive learner, $\mathcal{A}_{\mathrm{unif}}$, that uniformly samples its queries from a fixed uniform distribution, achieves a bounded expected error with a linear querying strategy.

   **Theorem 1.1** (Informal Version)**.** *Given an instance space $\mathcal{X}$ and a label space $\mathcal{Y}$, let $H \subseteq \mathcal{Y}^\mathcal{X}$ be a concept class where $LD(H)$ represents the Littlestone dimension of $H$. For*

*any $H$ that has $LD(H) < \infty$, $\mathcal{A}_{\mathrm{unif}}$ achieves an expected error bound $MB_{\mathcal{P}(H)}(\mathcal{A}_{\mathrm{unif}}) \leq \Delta LD(H)$ with a linear querying strategy $Q_{\mathcal{A}_{\mathrm{unif}}}(t) = O(t)$ where $\Delta$ is an input parameter.*

2. **Concept Class Learnability in the Blind-Prediction Setting.** Second, we show that non-trivial concept classes aren't learnable within the blind-prediction setting. Letting $H$ be any concept class that contains a classifier that labels two points differently, then any learning algorithm, adaptive or non-adaptive, is not learnable in the blind-prediction setting.

   **Theorem 1.2** (Informal Version). *For any $H$ and two points $x_1, x_2 \in \mathcal{X}$ such that $\exists h \in H$ where $h(x_1) \neq h(x_2)$, then for any learning algorithm $\mathcal{A}$, the expected mistake-bound $MB_{\mathcal{P}(H)}(\mathcal{A}) = \infty$.*

3. **Adaptive Learners for Pattern Classes.** As our third result, we investigate what types of learning algorithms are required to learn pattern classes under continuous data streams. In Section 4.3, we design a continuous pattern class $\mathcal{P}$, where each pattern $P \in \mathcal{P}$ is a continuous sequence of point-label pairs $(X_t, Y_t)_{t \geq 0}$, that is not learnable by any random sampling algorithm such as $\mathcal{A}_{\mathrm{unif}}$. Additionally, we construct an adaptive learning algorithm that successfully learns $\mathcal{P}$ with zero expected error. This important example signifies a learnability gap between concept classes and pattern classes.

4. **Discrete Data Streams.** Fourth, we develop a theory for realizable learning of pattern classes under discrete data streams in the blind-prediction setting for deterministic learning algorithms. We characterize a combinatorial quantity called the *query-learning distance* or $QLD$ for discrete pattern classes $\mathcal{P}$ with a query budget $Q \in \mathbb{N} \cup \{0\}$. We show that the optimal mistake-bound given $Q$ queries, $M_Q(\mathcal{P})$, is lower bounded by $QLD(\mathcal{P}, Q)$. Then, we construct a deterministic learning algorithm whose optimal mistake-bound is upper bounded by $QLD(\mathcal{P}, Q)$.

   **Theorem 1.3** (Informal Version). *For a discrete pattern class $\mathcal{P}$ and number of queries $Q$, the optimal mistake-bound $M_Q(\mathcal{P}) = QLD(\mathcal{P}, Q)$.*

## 1.4 Related Work

An extensively studied area in online learning theory closely related to our work is the round-by-round learning of concept classes from discrete data streams in the realizable setting. Littlestone [2] successfully characterized the types of concept classes $H$ that are learnable under an adversarial online setting which is now famously known as the Littlestone dimension or $LD(H)$. Later, Daniely et al. [3] extended this result to the multi-class setting, showing that $LD(H)$ also characterizes multi-class learnability. A recently explored setting called self-directed online learning shares an important trait with our learning frameworks which is adaptivity in selecting points where Devulapalli and Hanneke [4] constructed a dimension, $SDdim(H)$, characterizing learnable concept classes.

While traditional approaches assume that the learner receives the true label after each round, our study diverges by focusing on frameworks where feedback is only provided when actively queried by the learner. Our work is conceptually aligned with the area of partial monitoring, which investigates how various feedback constraints influence a learner's ability to minimize regret. A series of studies have established optimal regret bounds across different online learning scenarios, structured as discrete data streams with diverse feedback mechanisms [5–11].

A core principle within our learning frameworks is the ability of a learning algorithm to selectively query at different time-steps within a data-stream which is shared by stream-based active learning approaches. Several works within the field have explored theoretical guarantees of active learning in different variations of the stream-based setting [12–15]. However, a crucial difference between stream-based active learning and learning models in this work is the decision to query at a particular time is carried out before the current instance is observed.

## 2 Learning Frameworks

### 2.1 Basic Definitions

Let $\mathcal{X}$ and $\mathcal{Y}$ be arbitrary, non-empty sets where $\mathcal{X}$ is referred to as the instance space and $\mathcal{Y}$ is the label space. A concept class $H \subseteq \mathcal{Y}^{\mathcal{X}}$ consists of functions $f : \mathcal{X} \to \mathcal{Y}$. Depending on the context,

we will specify if we are considering a multi-class setting where $|\mathcal{Y}| \geq 2$ or a binary classification setting where $\mathcal{Y} = \{0, 1\}$.

To define a continuous data stream, we use the notation $(Z_t)_{t \geq 0} = (X_t, Y_t)_{t \geq 0}$ to define a point and label pair $Z_t = (X_t, Y_t)$ for each $t \in \mathbb{R}_{\geq 0}$. A continuous pattern class $\mathcal{P}$ is defined as $\mathcal{P} \subseteq \mathcal{C}((X_t, Y_t)_{t \geq 0})$ where $\mathcal{C}((X_t, Y_t)_{t \geq 0})$ represents the collection of all measurable continuous-time processes on the space $\mathcal{X} \times \mathcal{Y}$. Each pattern $P \in \mathcal{P}$ is then a continuous-time process $(Z_t)_{t \geq 0}$.

We now proceed to define discrete pattern classes and subsequently, discrete data streams. Let $\mathcal{Z} = \mathcal{X} \times \mathcal{Y}$ where $z \in \mathcal{Z}$ and $z = (x, y)$. Define $Z^\infty = (\mathcal{X} \times \mathcal{Y})^\infty$ which is the set of all countably infinite patterns. Then the discrete pattern class $\mathcal{P} \subseteq Z^\infty$. Both continuous and discrete pattern classes are referred to as $\mathcal{P}$ so it will be clear from context which type of pattern class we are referring to. It then follows that a discrete data stream $(Z_t)_{t=1}^\infty = (X_t, Y_t)_{t=1}^\infty$ lives in the space $\mathcal{Z}^\infty$.

## 2.2 Update-and-Deploy Setting

In this learning framework, we aim to describe the online learning game that occurs between a learner and an oblivious adversary. An oblivious adversary is an adversary impervious to any of the learner's actions; in other words, the adversary does not adapt its strategy based on the learner's actions. As a result, the oblivious adversary fixes the entire data stream in advance of the learning process.

Denote by $\mathcal{F}$ a class of predictor functions $\hat{f}$. With $\mathcal{D}$ representing the timestamps of the data stream, either discrete or continuous, then $\hat{f} : \mathcal{X} \times \mathcal{D} \to \mathcal{Y}$ is designed to make a prediction at every timestamp $t \in \mathcal{D}$. In the update-and-deploy setting, we consider the learning algorithm $\mathcal{A}$ and the predictor $\hat{f}$ to be separate entities. Denote by $Q_\mathcal{A}(t) = \{(X_{t_1}, Y_{t_1}), (X_{t_2}, Y_{t_2}), ...\}$ the set of queries made by learning algorithm $\mathcal{A}$ before time $t$. Intuitively, a learning algorithm is a mapping $\mathcal{A} : (\mathcal{X} \times \mathcal{Y})^* \to \mathcal{F}$ where $(\mathcal{X} \times \mathcal{Y})^*$ corresponds to the set $Q_\mathcal{A}(t)$. Formally, $\mathcal{A}((X_{t_1}, Y_{t_1}), ..., (X_{Q_\mathcal{A}(t)}, Y_{Q_\mathcal{A}(t)}))$ outputs a predictor $\hat{f} \in \mathcal{F}$ given the history of previous queries $Q_\mathcal{A}(t)$. It's important to note that we only consider learning algorithms $\mathcal{A}$ that have a linear querying strategy or $Q_\mathcal{A}(t) = O(t)$.

Assume the adversary has selected a data stream $(Z_t)_{t \in D}$. For each $t \in D$, the predictor $\hat{f}$ produces predictions $\hat{Y}_t = \hat{f}(X_t, t)$ given $X_t$ and $t$. On timestamps $t \in \mathcal{D}$ that the learning algorithm $\mathcal{A}$ decides to query, the following procedure occurs:

1. The learner $\mathcal{A}$ makes a decision to query and receives the true point-label pair $(X_t, Y_t)$.

2. $\mathcal{A}$ updates the predictor $\hat{f}$ with $(X_t, Y_t)$.

3. $\hat{f}$ is redeployed as the new predictor.

It's important to note that the data stream selected by the adversary is constrained to be realizable. If the realizability is with respect to a concept class $H$, then $\exists h \in H, \forall t \in D, h(X_t) = Y_t$. If the setting is studied under a discrete pattern class $\mathcal{P}$, then the pattern is considered realizable if $(X_t, Y_t)_{t=1}^\infty \in \mathcal{P}$. If $\mathcal{D}$ represents a continuous data stream and $\mathcal{P}$ a continuous pattern class, then the pattern $(X_t, Y_t)_{t \geq 0} \in \mathcal{P}$ implies realizability.

## 2.3 Blind-Prediction Setting

For our second learning framework, we describe the online learning game between the learner and an oblivious adversary. As similarly described in Section 2.2. an oblivious adversary acts independently of the learner's actions and fixes the entire data stream beforehand.

Let $\mathcal{A}$ be any learning algorithm and let $Q_\mathcal{A}(t)$ be the set of queries made by a learning algorithm $\mathcal{A}$ before time $t$. As mentioned in Section 2.2, we consider algorithms with a linear querying strategy where $Q_\mathcal{A}(t) = O(t)$. Letting $\mathcal{D}$ be the timestamps of the data stream, $\mathcal{A}$ is described as a mapping $\mathcal{A} : (\mathcal{X} \times \mathcal{Y})^* \times \mathcal{D} \to \mathcal{Y}$ where $(\mathcal{X} \times \mathcal{Y})^*$ corresponds to the set $Q_\mathcal{A}(t)$. At any time $t$, $\mathcal{A}$ only observes the current timestamp $t$ and the history of queries $Q_\mathcal{A}(t)$ when making a prediction $\hat{Y}_t$. If it decides to query, then $\mathcal{A}$ witnesses the true instance-label pair $(X_t, Y_t)$.

Assume that the adversary has selected a data stream $(Z_t)_{t \in D}$. For each $t \in \mathcal{D}$:

1. The learner $\mathcal{A}$ selects a prediction $\hat{Y}_t \in \mathcal{Y}$.

2. If the learner decided to query, then the pair $(X_t, Y_t)$ is revealed to the learner.

It's important to note that the data stream selected by the adversary is constrained to be realizable. Refer to Section 2.2 for realizability regarding concept classes and pattern classes.

## 2.4 Integral Mistake-Bounds

To capture the optimal behavior of learning algorithms under continuous data streams, we formalize the notion of integral mistake-bounds. Since we consider two separate settings, we construct a general mistake-bound and then differentiate from context which setting the mistake-bound operates under.

Due to their nature, pattern classes subsume concept classes so we define all the mistake-bounds with respect to pattern classes. Let $\mathcal{D} = \mathbb{R}_{\geq 0}$ which denotes the timestamps of a continuous stream. The pattern class representation of a concept class $H$, or $\mathcal{P}(H)$, is defined in the following way: $\mathcal{P}(H) = \{(X_t, Y_t)_{t \geq 0} : \exists h \in H, \forall t \in \mathcal{D}, h(X_t) = Y_t\}$. It is important to note that we assume that each pattern $P \in \mathcal{P}$ for any continuous pattern class $\mathcal{P}$ is measurable.

Given a continuous pattern class $\mathcal{P}$, a learning algorithm $\mathcal{A}$, and some realizable continuous data stream $(Z_t)_{t \geq 0}$, the quantity $MB_{\mathcal{P}}(\mathcal{A}, (Z_t)_{t \geq 0})$ represents the expected error $\mathcal{A}$ makes on the data stream $(Z_t)_{t \geq 0}$ given $\mathcal{P}$. Formally,

$$MB_{\mathcal{P}}(\mathcal{A}, (Z_t)_{t \geq 0}) = \lim_{T \to \infty} \mathbb{E}\left[\int_0^T \mathbb{1}[\mathcal{A}(X_t) \neq Y_t]\, dt\right].$$

To define the optimal mistake-bound for $\mathcal{P}$, we take the supremum over all patterns in the class:

$$MB_{\mathcal{P}}(\mathcal{A}) = \sup_{(Z_t)_{t \geq 0} = P \in \mathcal{P}} MB_{\mathcal{P}}(\mathcal{A}, (Z_t)_{t \geq 0}).$$

Finally, we obtain the optimal mistake-bound for the pattern class $\mathcal{P}$ by taking the infimum over all learning algorithms corresponding to the learning setting (blind-prediction or update-and-deploy):

$$MB_{\mathcal{P}} = \inf_{\mathcal{A}} MB_{\mathcal{P}}(\mathcal{A}).$$

## 3 Update-and-Deploy Setting: Learning Concept Classes from Continuous Data Streams

### 3.1 Littlestone Classes are Learnable

In this section, we are interested in multi-class concept classes $H$ that are learnable in the update-and-deploy setting with learning algorithms deploying a linear querying strategy. Below, we give a definition of the learnability of a concept class $H$ which allows us to frame our first important question.

**Definition 3.1.** *A concept class $H$ is learnable if the following condition is satisfied: there exists an algorithm $\mathcal{A}$ such that $MB_{\mathcal{P}(H)}(\mathcal{A}) < \infty$ and $Q_{\mathcal{A}}(t) = O(t)$.*

> **Question:** What is the dimension that characterizes the learnability of a concept class $H$ where finiteness implies learnability and an infinite value implies non-learnability?

Once we have defined learnability of a concept class $H$, our interest immediately swings towards the performance of different learning algorithms with linear querying strategies. Naturally, we want to understand if there exists optimal learning algorithms whose expected error is finite if the concept class $H$ is learnable. This then leads us to our second important question.

> **Question:** Does there exist a learning algorithm $\mathcal{A}$ employing a linear querying strategy such that for every $H$ that is learnable, does $MB_{\mathcal{P}(H)}(\mathcal{A}) < \infty$? If so, does the learning algorithm employ an adaptive strategy?

The Littlestone dimension [2] is a key measure that defines the learnability across various online learning frameworks. Extending this concept, we investigate whether the Littlestone dimension can similarly influence learnability in the context of continuous data streams. We propose that $LD(H)$

could be a valuable combinatorial tool for designing learning algorithms in the continuous setting. To explore this, we introduce Algorithm 1, or $\mathcal{A}_{\text{unif}}$, which is designed to learn any concept class with a finite Littlestone dimension, $LD(H) < \infty$, by using a linear querying approach.

The idea behind $\mathcal{A}_{\text{unif}}$ is to randomize the timestamp of the query so that the adversary has to "guess" which point in the data stream the learner will decide to target. If the timestamp of the query is not randomized, then the adversary can select a data stream designed with this knowledge. A potential strategy an adversary could employ against a deterministic learning algorithm would be to present the same point again and again to the learner for every query. Since the learner has only received information about one point, the adversary can present other points in the data stream at times the learner doesn't query forcing errors to occur. As a result, the adversary has a strategy to force an infinite mistake-bound to a learning algorithm that employs a deterministic querying strategy regardless if it's adaptive or non-adaptive.

To avoid this issue, we fitted $\mathcal{A}_{\text{unif}}$ with a randomized querying strategy. As shown in Algorithm 1, $\mathcal{A}_{\text{unif}}$ samples the next timestamp of the query, $t_q$, from a uniform distribution over an interval of fixed width $\Delta$.

---
**Algorithm 1** Uniform Sampler$(H, \Delta)$

---
**Require:** $H \neq \emptyset$
**Require:** $\Delta > 0$
1: $V = H, t = \text{time, starts at } t = 0, t_q \sim \text{Unif}[t, t + \Delta]$
2: Deploy $\hat{f}(x_t, t) = \arg\max_{r \in \{0,1\}} LD(V_{(x_t, r)})$
3: **while** true **do**
4:    **if** $t = t_q$ **then**
5:       Query at time $t_q$ and receive point-label pair $(x_{t_q}, y_{t_q})$
6:       Update $V = V_{(x_{t_q}, y_{t_q})}$
7:       Redeploy $\hat{f}(x_t, t) = \arg\max_{r \in \{0,1\}} LD(V_{(x_t, r)})$
8:       $t_q \sim \text{Unif}[t, t + \Delta]$
9:    **end if**
10: **end while**

---

In Algorithm 1, notice that the predictor function $\hat{f}$ follows that of the Standard Optimal Algorithm, or SOA, defined by Littlestone [2]. Since the $LD(H) < \infty$, and if the prediction differs from the true label on a query point, then the Littlestone dimension of the subsequent version space is reduced by at least 1. This property follows immediately from the analysis of the SOA, so the learner knows that it needs only $LD(H)$ successful queries to fully learn $H$ from the continuous data stream.

Additionally, note that while Algorithm 1 decides the next $t_q$ after the previous query finishes, this is done non-adaptively. The timestamp $t_q$ is not dependent on the true label witnessed by the previous queries; it's simply sampled from a uniform distribution. As a result, the set of query timestamps are produced in a non-adaptive fashion by sampling the next query from an interval of width $\Delta$.

**Theorem 3.2.** *Let $\mathcal{A}_{\text{unif}}$ be Algorithm 1. For any $H$ that has $LD(H) < \infty$, $MB_{\mathcal{P}(H)}(\mathcal{A}_{\text{unif}}) \leq \Delta LD(H)$ where $\Delta$ is an input parameter from Algorithm 1. Since $Q_{\mathcal{A}_{\text{unif}}}(t) = O(t)$, then $H$ is learnable.*

*Proof.* For a given $H$ with $LD(H) < \infty$, we show that the expected mistake-bound of algorithm $\mathcal{A}_{\text{unif}}$ is bounded proportionally to the size of $LD(H)$ using a linear querying strategy. Since $\mathcal{A}_{\text{unif}}$ deploys the SOA as its predictor, then the mistake-bound of $\mathcal{A}_{\text{unif}}$ is inherently tied to $LD(H)$. In other words, if $\mathcal{A}_{\text{unif}}$ makes $LD(H)$ successful queries, where success implies that the SOA's prediction is incorrect on the query point, then the version space has Littlestone dimension of $0$ implying that any consistent classifier subsequently makes zero error onwards. Our analysis first focuses on bounding the maximum expected error $\mathcal{A}_{\text{unif}}$ makes until its first successful query. We repeat this analysis $LD(H) - 1$ times to show that $MB_{\mathcal{P}(H)}(\mathcal{A}_{\text{unif}}) \leq \Delta LD(H)$ with a linear querying strategy.

As a starting point, we define all the necessary quantities in order to begin the analysis. Since our learning model assumes an oblivious adversary, it selects a continuous data stream $(Z_t)_{t \geq 0}$ realizable with respect to some target concept $f^* \in H$ before the learning process begins. Let the random

variable $B_k$ be an indicator random variable representing the success of the $k^{th}$ query on the process $(Z_t)_{t \geq 0}$. More specifically,

$$B_k = \begin{cases} 1 & \text{if the } k^{th} \text{ query is successful} \\ 0 & \text{else} \end{cases}$$

takes a value of 1 if the $k^{th}$ query succeeds. Then, we define $P(B_k = 1 | B_{k-1} = 0, B_{k-2} = 0, ..., B_1 = 0) = \epsilon_k$ which is the probability that the learner has a successful query on the $k^{th}$ try given that the previous $k - 1$ attempts failed. $\epsilon_k$ can be equivalently viewed as the probability of the learner making an error on the $k^{th}$ interval because a successful query results in receiving a mistake-point, or a point the predictor incorrectly predicts. Since $\mathcal{A}_{\text{unif}}$ selects its $k^{th}$ query $t_q^k$ from a $\Delta$-sized interval, then $\Delta \epsilon_k$ represents the total potential error the learner makes on the $k^{th}$ interval.

Our primary interest is calculating the expected error $\mathcal{A}_{\text{unif}}$ makes until it reaches $LD(H)$ successful queries. Since the learner $\mathcal{A}_{\text{unif}}$ deploys an SOA predictor, $LD(H)$ successful queries where the predictor is incorrect guarantees the learner to narrow down on the right set of consistent classifiers.

We approach this by first computing the expected error that the learning algorithm makes until its first successful query. It's important to note that $\mathcal{A}_{\text{unif}}$ does not alter its querying strategy regardless of the number of successful queries it has received; it constantly chooses its queries from intervals of size $\Delta$. As a result, after the learner receives its first successful query, the same process repeats again until $\mathcal{A}_{\text{unif}}$ finds it second successful query. So, we focus on bounding the maximum expected error $\mathcal{A}_{\text{unif}}$ will encounter until its next successful query for the data stream $(Z_t)_{t \geq 0}$.

Let $A$ be a function that represents the maximum error the learner receives until its first successful query given the values of the random variables $B_1, B_2, ...$ Formally speaking, let $A = A(B_1, B_2, ...) = \Delta(\epsilon_1 + \epsilon_2(1 - B_1) + \epsilon_3(1 - B_1)(1 - B_2) + \cdots) = \Delta \sum_{k=1}^{\infty} \epsilon_k \Pi_{i=1}^{k-1}(1 - B_i)$. Each $\Delta \epsilon_k$ represents the error region in the $k^{th}$ interval given that the previous $k - 1$ queries failed or each $B_i = 0$ for all $i \leq k - 1$. It's important to observe that $A$ is the maximum error the learner receives until the first successful query. As an example, let $B_n = 1$ for some $n \in \mathbb{N}$ and $B_j = 0$ for all $j < n$. Then $A$ includes the cumulative error from the first $n - 1$ intervals and the entire potential error on the $n^{th}$ interval (represented as $\Delta \epsilon_n$) even though the $n^{th}$ query, which is successful, can lie anywhere within the $\Delta \epsilon_n$ error region located inside the $n^{th}$ interval. Now, we compute the expectation of $A$.

$$\mathbb{E}[A] = \mathbb{E}\left[\Delta \sum_{i=1}^{\infty} \epsilon_k \Pi_{i=1}^{k-1}(1 - B_i)\right] = \Delta \sum_{k=1}^{\infty} \epsilon_k \mathbb{E}\left[\Pi_{i=1}^{k-1}(1 - B_i)\right]$$

$$= \Delta \sum_{k=1}^{\infty} \epsilon_k P(B_1 = 0, ..., B_{k-1} = 0) = \Delta \sum_{k=1}^{\infty} \epsilon_k \Pi_{i=1}^{k-1}(1 - \epsilon_i)$$

Since we are interested in the maximum expected error the learner $\mathcal{A}_{\text{unif}}$ encounters until its first successful query, we want to bound the term $\Delta \sum_{k=1}^{\infty} \epsilon(k) \Pi_{i=1}^{k-1}(1 - \epsilon_i)$ by selecting the optimal values for $\epsilon_1, \epsilon_2, ...$ Notice that the expression is recursive in the sense that if we pulled out the first $k$ terms, the structure of the sum doesn't change. We then exploit this fact to bound the total value of the sum. Let $U^* = \sup_{\vec{\epsilon} \in [0,1]^\infty} \Delta \sum_{k=1}^{\infty} \vec{\epsilon}(k) \Pi_{i=1}^{k-1}(1 - \vec{\epsilon}(i))$ where $\vec{\epsilon}(1) = \epsilon_1, \vec{\epsilon}(2) = \epsilon_2$, and so on and so forth. Then,

$$U^* = \sup_{\vec{\epsilon} \in [0,1]^\infty} \Delta \sum_{k=1}^{\infty} \vec{\epsilon}(k) \Pi_{i=1}^{k-1}(1 - \vec{\epsilon}(i)) = \sup_{\vec{\epsilon} \in [0,1]^\infty} \Delta \vec{\epsilon}(1) + \Delta(1 - \vec{\epsilon}(1)) \sum_{k=2}^{\infty} \vec{\epsilon}(k) \Pi_{i=2}^{k-1}(1 - \vec{\epsilon}(i))$$

$$\leq \sup_{p \in [0,1]} \Delta p + (1 - p) \left( \sup_{\vec{\epsilon} \in [0,1]^\infty} \Delta \sum_{k=1}^{\infty} \vec{\epsilon}(k) \Pi_{i=1}^{k-1}(1 - \vec{\epsilon}(i)) \right) \leq \sup_{p \in [0,1]} \Delta p + (1 - p) U^*$$

$$\leq \sup_{p \in [0,1]} \frac{\Delta p}{1 - (1 - p)} = \Delta.$$

Therefore, we show that $E[A] \leq \Delta$.

At the beginning of this analysis, we assumed some adversarially chosen data stream and target concept, so the result $E[A] \leq \Delta$ holds for any choice of $(Z_t)_{t \geq 0}$ realizable with respect to $H$.

Now, we repeat this analysis $LD(H) - 1$ times. Therefore, $MB_{\mathcal{P}(H)}(\mathcal{A}_{\text{unif}}) \leq \Delta LD(H)$ where $Q_{\mathcal{A}_{\text{unif}}}(t) = O(t)$.
□

In Theorem 3.2, we establish that if $LD(H)$ is finite, then $H$ is learnable in the update-and-deploy setting. This leads to our second result, which demonstrates that $LD(H)$ serves as the defining dimension for the learnability of a concept class $H$ in this context.

**Theorem 3.3.** *If $LD(H) = \infty$, then for any learning algorithm $\mathcal{A}$ with a linear querying strategy $Q_{\mathcal{A}}(t)$, $MB_{\mathcal{P}(H)}(\mathcal{A}) = \infty$ implying that $H$ is not learnable.*

For the formal proof of Theorem 3.3, refer to Appendix A.1. While the results hold for $O(t)$ querying strategies, an open direction is to investigate algorithms with a broader range of querying strategies.

# 4  Blind-Prediction Setting: Learning from Discrete and Continuous Data Streams

## 4.1  It's Impossible to Learn Non-Trivial Concept Classes from Continuous Data Streams

In this section, we discover what constitutes learnability of multi-class concept classes in the blind-prediction setting. We borrow Definition 3.1 to describe the learnability of a concept class $H$.

Since the blind-prediction setting is a harder variant of the update-and-deploy setting, we frame a similar question asking if the Littlestone dimension is the right characterization of learnability.

**Question:** What characterizes the learnability of concept classes $H$ in the blind-prediction setting?
Does $LD(H)$ play a pertinent role?

To answer this question, we come up with a simple concept class $H$ that proves to be unlearnable in the blind-prediction setting. This result comes in stark contrast to the results found in Section 3.1. Below, we detail Theorem 4.1 and Corollary 4.2.

**Theorem 4.1.** *Let $H = \{h\}$ and $\mathcal{X} = \{x_1, x_2\}$ with $h(x_1) = 0$ and $h(x_2) = 1$. Then, for any learning algorithm $\mathcal{A}$ with a linear querying strategy, $MB_{\mathcal{P}(H)}(\mathcal{A}) = \infty$ so $H$ is not learnable under the blind-prediction setting.*

For the formal proof of Theorem 4.1, refer to Appendix A.2.

**Corollary 4.2.** *If $H$ is a concept class such that $\exists h \in H$ and $\exists x_1, x_2 \in \mathcal{X}$ such that $h(x_1) \neq h(x_2)$, then $H$ is unlearnable in the blind-prediction setting.*

*Proof.* Let $H' = h$ and $\mathcal{X}' = \{x_1, x_2\}$. From Theorem 4.1, it was shown that $MB_{\mathcal{P}(H')}(\mathcal{A}) = \infty$ for any learning algorithm $\mathcal{A}$ with a linear querying strategy so $H'$ is unlearnable. Since $H' \subseteq H$ and $x_1, x_2 \in \mathcal{X}$, it follows that $MB_{\mathcal{P}(H)} = \infty$ so $H$ is unlearnable.
□

## 4.2  Are Adaptive Learners Required for Pattern Classes?

In this section, we demonstrate the necessity of adaptive learning algorithms for effectively learning pattern classes. Since concept classes represent a set of functions, and functions can be thought as established input-output pairs, different permutations of these pairs don't result in different functions being realizable on the sequence. As a result, non-adaptive learning algorithms are sufficient in learning concept classes but adaptive learning strategies may be required for pattern classes. Below, we construct an example of a continuous pattern class $\mathcal{P}$ that is only learnable by any adaptive sampling algorithm.

**Pattern Class Example**  Let $H$ be a multi-class concept class with $LD(H) = \infty$. For $t_1, t_2 \in \mathbb{N}$ with $t_2 > t_1$, define $\bar{\mathcal{P}}(H, t_1, t_2) = \{(X_t, Y_t)_{t \in (t_1, t_2)} : \exists h \in H, \forall t \in (t_1, t_2), h(X_t) = Y_t\}$. Then, $\mathcal{P}(H, t_1, t_2) = \{(X_t, Y_t)_{t \in [t_1, t_2)} : \exists P \in \bar{P}(H, t_1, t_2) \text{ such that } (X_{t_1}, Y_{t_1}) = (P, t_2) \text{ and } (X_t, Y_t)_{t \in (t_1, t_2)} = P\}$. $\bar{\mathcal{P}}(H, t_1, t_2)$ corresponds to the set of realizable data streams between $t_1$ and $t_2$ and $\mathcal{P}(H, t_1, t_2)$ ensures that at time $t_1$ the data stream encodes the entire pattern from $t_1$ to $t_2$ in $X_{t_1}$. Let $\mathcal{N} = \{\mathbf{n} \in \{0\} \times \mathbb{N}^\infty : \forall i \in \mathbb{N}, \mathbf{n}(i+1) > \mathbf{n}(i)\}$ and

$\mathcal{Q} = \{\mathbf{q} \in \{0\} \times \mathbb{Q}_{>0}^{\infty} : \exists \mathbf{n} \in \mathcal{N}, \forall i \in \mathbb{N}, \mathbf{n}(i) < \mathbf{q}(i+1) < \mathbf{n}(i+1)\}$. Then, we define the continuous pattern class $\mathcal{P}$ in the following way: $\mathcal{P} = \bigcup_{\mathbf{q} \in \mathcal{Q}} \left( \prod_{i=1}^{\infty} \mathcal{P}(H, \mathbf{q}(i), \mathbf{q}(i+1)) \right)$ where $\prod_{i=1}^{\infty} \mathcal{P}(H, \mathbf{q}(i), \mathbf{q}(i+1))$ represents an infinite Cartesian product among the valid patterns in each interval dictated by $\mathbf{q}$.

**Lemma 4.3.** *For the update-and-deploy setting, any random sampling algorithm $\mathcal{A}$ with a linear querying strategy $Q_{\mathcal{A}}(t)$ has $MB_{\mathcal{P}}(\mathcal{A}) = \infty$.*

*Proof.* Let $\mathcal{A}$ be a random sampling algorithm with a linear querying strategy $Q_{\mathcal{A}}(t)$. We will now construct a continuous data stream $(Z_t)_{t \geq 0}$ that is realizable with respect to $\mathcal{P}$ in a randomized fashion and prove a bound on the minimum expected error. Randomly select a vector $\mathbf{q} \in \mathcal{Q}$.

To construct such a continuous process $(Z_t)_{t \geq 0}$, we first decompose $\mathbb{R}_{\geq 0} = \cup_{n=1}^{\infty} [2\mathbf{q}(n), 2\mathbf{q}(n+1))$. The idea behind this decomposition is to construct a pattern on each interval that corresponds to a randomly chosen $h \in H$. To do this, we take each interval $[2\mathbf{q}(n), 2\mathbf{q}(n + 1))$, letting $\mathcal{Q}_{\mathcal{A}}(2\mathbf{q}(n + 1)) = k$ for some $k \in \mathbb{N}$, and break it further down such that $[2\mathbf{q}(n), 2\mathbf{q}(n + 1)) = \cup_{j=1}^{2k} \left[ \frac{(\mathbf{q}(n+1) - \mathbf{q}(n))}{k}(j-1) + 2\mathbf{q}(n), \frac{(\mathbf{q}(n+1) - \mathbf{q}(n))}{k}j + 2\mathbf{q}(n) \right)$. By doing this, we can take an arbitrary root-to-leaf path from a Littlestone tree of depth $2k$, and then paint each sub-interval with an instance-label pair on this path. Since the number of sub-intervals is greater than the number of queries made by the algorithm $\mathcal{A}$, on some set of sub-intervals the algorithm $\mathcal{A}$ is forced to guess the true label.

As described above, assume the interval $[2\mathbf{q}(n), 2\mathbf{q}(n+1))$ for some $n \in \mathbb{N}$, letting $\mathcal{Q}_{\mathcal{A}}(2\mathbf{q}(n+1)) = k$ for some $k \in \mathbb{N}$. Since $LD(H) = \infty$, there must exist a Littlestone tree $T$ where the minimum root-to-leaf depth is at least $2k$. Then, let $\sigma = \{(X_1, Y_1), ..., (X_{2k}, Y_{2k})\}$ correspond to a randomly chosen root-to-leaf path. For each $j \in \{1, ..., 2k\}$, populate the interval $\left[ \frac{(\mathbf{q}(n+1) - \mathbf{q}(n))}{k}(j-1) + 2\mathbf{q}(n), \frac{(\mathbf{q}(n+1) - \mathbf{q}(n))}{k}j + 2\mathbf{q}(n) \right)$ with the pair $(X_j, Y_j)$. For simplicity, let $I_j = \left[ \frac{(\mathbf{q}(n+1) - \mathbf{q}(n))}{k}(j-1) + 2\mathbf{q}(n), \frac{(\mathbf{q}(n+1) - \mathbf{q}(n))}{k}j + 2\mathbf{q}(n) \right)$. For the process at time $t = 2\mathbf{q}(n)$, let $Z_t = (P, 2\mathbf{q}(n + 1))$ where $P = (X_1, Y_1)_{t \in I_1 \setminus 2\mathbf{q}(n)} \cup \bigcup_{j=2}^{2k} [X_j, Y_j)_{t \in I_j}$.

It's important to note that the constructed continuous process on the interval $[2\mathbf{q}(n), 2\mathbf{q}(n + 1))$ lies in $\mathcal{P}(H, 2\mathbf{q}(n), 2\mathbf{q}(n + 1))$. The first point in the interval corresponds to the point $(P, 2\mathbf{q}(n + 1))$ and $P \in \mathcal{P}(H, 2\mathbf{q}(n), 2\mathbf{q}(n + 1))$ since it was generated from a root-to-leaf path in $T$ which is realizable by some $h \in H$.

Now, we show that on the sub-intervals $\mathcal{A}$ does not query in the interval $[2\mathbf{q}(n), 2\mathbf{q}(n + 1))$, $\mathcal{A}$, the minimum expected error is equal to $\frac{(\mathbf{q}(n+1) - \mathbf{q}(n))}{k}$. The analysis closely mirrors that of in Theorem 3.3. Let the $j^{th}$ sub-interval be a sub-interval, where $1 \leq j \leq 2k$, where $\mathcal{A}$ does not query. Let $E_1 = \{t \in I_j : \mathcal{A}(X_t) \neq Y_j\}$ and $E_0 = \{t \in I_j : \mathcal{A}(X_t) \neq Y'_j\}$ where $Y'_j$ is the other label in tree $T$ for the point $X_j$. Then, $\mathbb{E}\left[ \int_{I_j} \mathbb{1}[\mathcal{A}(X_t) \neq Y_t] \, dt \right] = \mathbb{E}\left[ \mu(E_1)\mathbb{1}[Y_t = Y_j] + \mu(E_0)\mathbb{1}[Y_t = Y'_j] \right] = \mathbb{E}[\mu(E_1)]\mathbb{E}[\mathbb{1}[Y_t = Y_j]] + \mathbb{E}[\mu(E_0)]\mathbb{E}[\mathbb{1}[Y_t = Y'_j]]$ where $\mu$ is the Lebesgue measure. Since a random branch was chosen within the tree $T$, there was an equal chance of selecting $Y_t = Y_j$ or $Y_t = Y'_j$, then $\mathbb{E}[\mu(E_1)]\mathbb{E}[\mathbb{1}[Y_t = Y_j]] + \mathbb{E}[\mu(E_0)]\mathbb{E}[\mathbb{1}[Y_t = Y'_j]] = \mathbb{E}[\mu(E_0)]/2 + \mathbb{E}[\mu(E_1)]/2 = \mathbb{E}[\mu(E_0) + \mu(E_1)]/2 \geq \frac{(\mathbf{q}(n+1) - \mathbf{q}(n))}{k}$. Therefore, the learner $\mathcal{A}$ accumulates an expected error of $\frac{(\mathbf{q}(n+1) - \mathbf{q}(n))}{k}$ on each interval it doesn't query. Since the learner has only $k$ queries, it can only query in at most $k$ of the $2k$ intervals. At minimum there will exist $k$ intervals that haven't been queried by the learner. Let $I_1, ..., I_k$ represent $k$ of these intervals algorithm $\mathcal{A}$ does not query. It follows that $\mathbb{E}\left[ \int_{2\mathbf{q}(n)}^{2\mathbf{q}(n+1)} \mathbb{1}[\mathcal{A}(X_t) \neq Y_t] \, dt \right] \geq \sum_{i=1}^{k} \mathbb{E}\left[ \int_{I_i} \mathbb{1}[\mathcal{A}(X_t) \neq Y_t] \, dt \right] = \mathbf{q}(n + 1) - \mathbf{q}(n)$.

Since $n \in \mathbb{N}$ was chosen arbitrarily, then it holds for all intervals $[2\mathbf{q}(n), 2\mathbf{q}(n + 1))$. As a result,

$$\lim_{T \to \infty} \mathbb{E}\left[ \int_0^T \mathbb{1}[\mathcal{A}(X_t) \neq Y_t] \, dt \right] \geq \lim_{T \to \infty} \sum_{i=1}^{\max\{n \in \mathbb{N}: T \geq 2\mathbf{q}(n+1)\}} \mathbb{E}\left[ \int_{2\mathbf{q}(i)}^{2\mathbf{q}(i+1)} \mathbb{1}[\mathcal{A}(X_t) \neq Y_t] \, dt \right]$$

$$= \lim_{T \to \infty} \sum_{i=1}^{\max\{n \in \mathbb{N}: T \geq 2\mathbf{q}(n+1)\}} \mathbf{q}(i + 1) - \mathbf{q}(i) = \infty.$$

Since we constructed the process $(Z_t)_{t \geq 0}$ by randomly selecting branches from Littlestone trees for each interval, we appeal to the probabilistic method to show that there exists a fixed choice of a continuous-process $(Z_t)_{t \geq 0}$ such that $MB_\mathcal{P}(\mathcal{A}, (Z_t)_{t \geq 0}) = \infty$. Therefore, $MB_\mathcal{P}(\mathcal{A}) = \infty$. $\qquad \square$

**Remark 4.4.** *It can be shown that the results of Lemma 4.3 directly extend for the blind-prediction setting.*

Now, we turn to an adaptive sampling learning algorithm, specifically Algorithm 2, that achieves $MB_\mathcal{P}(\text{Algorithm 2}) = 0$. Specifically, we show this result in the blind-prediction setting. As will be proven in the analysis of Lemma 4.5, Algorithm 2 specifically queries at the timestamps where a portion of the future continuous data stream is revealed. As a result, Algorithm 2 makes at most a countable number of mistakes because only a countable number of such points exist in any continuous data stream realizable by $\mathcal{P}$ so it has an expected error of 0 with a linear querying strategy.

---

**Algorithm 2** Adaptive Sampler($\mathcal{P}$)

---

1: $t = $ time {starts at $t = 0$}, $t_q = 0$, initialize $\hat{f}$ to be some function $\hat{f} : \mathbb{R}_{\geq 0} \to \mathcal{Y}$
2: **while** true **do**
3:     Predict $\hat{f}(t)$
4:     **if** $t = t_q$ **then**
5:         Query and receive point-label pair $(X_t, Y_t) = (P, n)$
6:         Update $\hat{f}(t) = Y_t$ for all $(X_t, Y_t) \in P$
7:         $t_q \leftarrow n$
8:     **end if**
9: **end while**

---

**Lemma 4.5.** *Let $\mathcal{A}$ be the adaptive sampler in Algorithm 2. Then, the querying strategy $Q_\mathcal{A}(t) \leq t$ and $MB_\mathcal{P}(\mathcal{A}) = 0$ in the blind-prediction setting.*

*Proof (Sketch).* Let $\mathcal{A}$ represent Algorithm 2 and $(Z_t)_{t \geq 0} = P \in \mathcal{P}$ be any adversarially chosen data stream. Since each $\mathbf{q} \in \mathcal{Q}$ has $\mathbf{q}(1) = 0$, then for any $P \in \mathcal{P}$, it must be the case that $Z_0 = (X_0, Y_0)$ where $X_0$ reveals the full sequence until time $Y_0$. Algorithm 2 has its first query at time $t = 0$ and fits the predictor $\hat{f}$ to output the labels of sequence $X_0$ for all time $t \in (0, Y_0)$. Since $(Z_t)_{t > 0}$ follows the exact sequence described by $X_0$ until time $t = Y_0$, then $\int_0^{Y_0} \mathbb{1}[\mathcal{A}(X_t) \neq Y_t] \, dt = 0$ implying that $\mathbb{E}[\int_0^{Y_0} \mathbb{1}[\mathcal{A}(X_t) \neq Y_t] \, dt] = 0$. At time $t = Y_0$, $\mathcal{A}$ queries at exactly the right time to gain information about a future portion of the data stream $(Z_t)_{t \geq 0}$ with the same analysis repeating continuously. As a result, $MB_\mathcal{P}(\mathcal{A}, (Z_t)_{t \geq 0}) = 0$ and since $(Z_t)_{t \geq 0}$ was arbitrarily chosen, then $MB_\mathcal{P}(\mathcal{A}) = 0$ with $Q_\mathcal{A}(t) \leq t$. $\qquad \square$

**Remark 4.6.** *The results of Lemma 4.5 can also be extended to the update-and-deploy setting.*

### 4.3 Learning Pattern Classes from Discrete Data Streams

As witnessed in the example from Section 4.2, one can construct a rather complex pattern class to model almost any sort of structure. While the power of pattern classes is inherent in their ability to express complicated relationships, directly analyzing their behavior under continuous data streams without a foundational understanding can prove to be an intractable problem.

As a result, we initiate a study of pattern classes under discrete data streams to provides a foundational understanding of how learning algorithms handle data arriving in distinct, separate chunks. This framework simplifies the complexity by allowing us to focus on key principles of sequential decision-making such as incremental learning. By developing a theory in a discrete context, we can potentially employ these insights that can prove to be crucial for tackling the more complex scenarios of learning under continuous data streams. In Appendix B, we develop a complete theory on realizable learning of pattern classes in the blind-prediction setting under discrete streams.

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

# A  Proofs for Learning from Continuous Data Streams

## A.1  Proof of Theorem 3.3

*Proof.* Let $\mathcal{A}$ be a learning algorithm with a linear querying strategy $Q_{\mathcal{A}}(t)$. Assume some concept class $H$ with $LD(H) = \infty$. For every $n \in \mathbb{N}$, we show that there exists an adversarially constructed data stream, $(Z_t)_{t \geq 0}$, such that $MB_{\mathcal{P}(H)}(\mathcal{A}, (Z_t)_{t \geq 0}) \geq n$. Since this holds $\forall n \in \mathbb{N}$, then $MB_{\mathcal{P}(H)}(\mathcal{A}) = \sup_{(Z_t)_{t \geq 0}} MB_{\mathcal{P}(H)}(\mathcal{A}, (Z_t)_{t \geq 0}) = \infty$.

Our learning model assumes an oblivious adversary, so we will construct a continuous data stream $(Z_t)_{t \geq 0}$ beforehand that is realizable with respect to $H$. However, we construct $(Z_t)_{t \geq 0}$ in a randomized fashion and bound the minimum expected error of this randomly constructed process. Then, at the end of the proof, we will call upon the probabilistic method to show that there exists a continuous process achieving at least that expected error.

Take some $n \in \mathbb{N}$ and let $Q_{\mathcal{A}}(4n) = k$. Since $LD(H) = \infty$, then there must exist a Littlestone tree $T$ where the minimum root-to-leaf path of $T$ is at least $2k$. Consider a random walk in $T$ starting at the root node that picks with probability $1/2$ the left child or the right child and descends level-by-level until it reaches a leaf node. Let $\sigma = \{(x_1, y_1), ..., (x_{2k}, y_{2k})\}$ correspond to the root-to-leaf path produced by the random walk on $T$. Note that $H_\sigma = \{h \in H : \forall (X_i, Y_i) \in \sigma, h(X_i) = Y_i\}$, the subset of the concept class consistent with the sequence $\sigma$, will have at least one classifier due to the guarantee provided by the Littlestone tree that every branch in $T$ is realizable by some $h \in H$.

Now, we describe the construction of the continuous process $(Z_t)_{t \geq 0}$ using the sequence $\sigma$. Decompose $[0, 4n)$ in the following way: $[0, 4n) = \bigcup_{j=1}^{2k} \left[ \frac{2n}{k}(j-1), \frac{2n}{k}j \right)$. For the $j^{th}$ interval where $1 \leq j \leq 2k$, $\forall t \in \left[ \frac{2n}{k}(j-1), \frac{2n}{k}j \right)$, define $Z_t = (X_j, Y_j) = \sigma(j)$. On the time interval $[0, 4n)$, if one segments the process $(Z_t)_{t \geq 0}$ into intervals of size $\frac{2n}{k}$, then for the $j^{th}$ interval, where $1 \leq j \leq 2k$, the process contains the point $\sigma(j)$ for the entirety of the interval. For $t \geq 4n$, then define $Z_t = (X_t, Y_t)$ to be a point-label pair $(X', Y')$ such that for each $h \in H_\sigma, h(X') = Y'$.

Now, we show that on the intervals $\mathcal{A}$ does not query in the time period $[0, 4n)$, the minimum expected error is equal to $\frac{n}{k}$. Let the $j^{th}$ interval be an interval, where $1 \leq j \leq 2k$, where $\mathcal{A}$ does not query. Let $E_1 = \{t \in \left[ \frac{2n}{k}(j-1), \frac{2n}{k}j \right] : \mathcal{A}(X_t) \neq Y_j\}$ and $E_0 = \{t \in \left[ \frac{2n}{k}(j-1), \frac{2n}{k}j \right] : \mathcal{A}(X_t) \neq Y_j'\}$ where $Y_j'$ is the other label in tree $T$ for the point $X_j$. Then, $\mathbb{E}\left[ \int_{\frac{2n}{k}(i-1)}^{\frac{2n}{k}i} \mathbb{1}[\mathcal{A}(X_t) \neq Y_t] \, dt \right] = \mathbb{E}\left[ \mu(E_1) \mathbb{1}[Y_t = Y_j] + \mu(E_0) \mathbb{1}[Y_t = Y_j'] \right] = \mathbb{E}[\mu(E_1)] \mathbb{E}[\mathbb{1}[Y_t = Y_j]] + \mathbb{E}[\mu(E_0)] \mathbb{E}[\mathbb{1}[Y_t = Y_j']]$ where $\mu$ is the Lebesgue measure. Since a random branch was chosen within the tree $T$, there was an equal chance of selecting $Y_t = Y_j$ or $Y_t = Y_j'$, then $\mathbb{E}[\mu(E_1)] \mathbb{E}[\mathbb{1}[Y_t = Y_j]] + \mathbb{E}[\mu(E_0)] \mathbb{E}[\mathbb{1}[Y_t = Y_j']] = \mathbb{E}[\mu(E_0)]/2 + \mathbb{E}[\mu(E_1)]/2 = \mathbb{E}[\mu(E_0) + \mu(E_1)]/2 \geq n/k$. Therefore, the learner $\mathcal{A}$ accumulates an expected error of $n/k$ on each interval it doesn't query. Since the learner has only $k$ queries, it can only query in at most $k$ of the $2k$ intervals. At minimum there will exist $k$ intervals that haven't been queried by the learner. Let $I_1, ..., I_k$ represent $k$ of these intervals algorithm $\mathcal{A}$ does not query. It follows that $\mathbb{E}\left[ \int_0^{4n} \mathbb{1}[\mathcal{A}(X_t) \neq Y_t] \, dt \right] \geq \sum_{i=1}^k \mathbb{E}\left[ \int_{I_i} \mathbb{1}[\mathcal{A}(X_t) \neq Y_t] \, dt \right] = n$.

The strategy chosen to prove a lower bound on the expected error to be $n$ relied on generating a continuous time process by randomly selecting a branch within the Littlestone tree $T$ which corresponds to a random selection of a target concept. However, we appeal to the probabilistic method to show that if the expected error for algorithm $\mathcal{A}$ is at least $n$, then there exists a fixed choice of a continuous-process $(Z_t)_{t \geq 0}$ such that $MB_{\mathcal{P}(H)}(\mathcal{A}, (Z_t)_{t \geq 0}) \geq n$. Since we show that for every $n \in \mathbb{N}$ and any learning algorithm $\mathcal{A}$ there exists an adversarial strategy $(Z_t)_{t \geq 0}$ such that $MB_{\mathcal{P}(H)}(\mathcal{A}, (Z_t)_{t \geq 0}) \geq n$, then the adversary can force the learner to make an arbitrarily large error implying that $MB_{\mathcal{P}(H)}(\mathcal{A}) = \sup_{(Z_t)_{t \geq 0}} MB_{\mathcal{P}(H)}(\mathcal{A}, (Z_t)_{t \geq 0}) = \infty$ so $H$ is not learnable. $\square$

## A.2  Proof of Theorem 4.1

*Proof.* The essence of this proof lies in the simple yet effective scheme the adversary can employ to force any learning algorithm $\mathcal{A}$ with a linear querying strategy $Q_{\mathcal{A}}(t)$ to have $MB_{\mathcal{P}(H)}(\mathcal{A}) = \infty$. The idea behind this adversarial approach is to divide the timeline, $\mathbb{R}_{\geq 0}$, into small enough intervals

where each interval is populated randomly with point-label pair $(x_1, 0)$ or $(x_2, 1)$ such that $\mathcal{A}$ is forced to guess the right label.

We first describe the construction of the continuous data stream $(Z_t)_{t \geq 0}$ realizable with respect to $h$. As previously done in Theorem 3.3, we use a randomized approach in constructing $(Z_t)_{t \geq 0}$ to prove a bound on the expected error. This randomized approach draws upon a family of continuous processes to show that the expected error is some minimum value. However, the mistake-bounds require a singular continuous process to yield that error. So, we apply the probabilistic method to prove the existence of a continuous data stream that can be fixed beforehand that achieves at least that expected error.

We first describe the construction of $(Z_t)_{t \geq 0}$ in a randomized fashion. Decompose $\mathbb{R}_{\geq 0} = \cup_{n=1}^{\infty} [n - 1, n)$. For every $n \in \mathbb{N}$, let $k_n = Q_{\mathcal{A}}(n)$. Then, split $[n-1, n)$ into $2k_n$ intervals each of size $\frac{1}{2k_n}$ such that $[n - 1, n) = \bigcup_{j=1}^{2k_n} \left[ n - 1 + \frac{1}{2k_n}(j - 1), n - 1 + \frac{1}{2k_n} j \right)$. Let $\sigma \sim \text{Unif}(\{(x_1, 0), (x_2, 1)\}^{2k_n})$ be a sequence sampled uniformly from the space $\{(x_1, 0), (x_2, 1)\}^{2k_n}$. Then, construct the process $(Z_t)_{t \geq 0}$ such that $\forall n \in \mathbb{N}, \sigma \sim \text{Unif}(\{(x_1, 0), (x_2, 1)\}^{2k_n}), \forall j \in \{1, ..., 2k_n\}, \forall t \in [n - 1 + \frac{1}{2k_n}(j - 1), n - 1 + \frac{1}{2k_n} j)$ then $Z_t = (X_j, Y_j) = \sigma(j)$. Essentially, we assign the point-label pairs $(x_1, 0)$ and $(x_2, 1)$ randomly to each interval to construct the process.

Letting $n \in \mathbb{N}$, then algorithm $\mathcal{A}$ makes at most $k_n$ queries in the interval $[n - 1, n)$ implying at least $k_n$ of the intervals within $[n - 1, n)$ pass by the learner with no query. On the intervals the learner does not query, we show that the learner's expected error is equal to $\frac{1}{4k_n}$. For some $1 \leq j \leq 2k_n$, let the $j^{th}$ interval within $[n - 1, n)$ contain no queries from $\mathcal{A}$. Then, let $E_0$ and $E_1$ represent the portion of the $j^{th}$ interval that $\mathcal{A}$ predicts as a 0 or 1 respectively. It follows that the expected error of algorithm $\mathcal{A}$ on this interval is equivalent to $\mathbb{E}[\mu(E_1)\mathbb{1}[Y_j = 0] + \mu(E_0)\mathbb{1}[Y_j = 1]]$ where $Y_j$ is the true label for the $j^{th}$ interval and $\mu$ is the Lebesgue measure. Since there's an equal probability of $Y_j = 0$ or $Y_j = 1$, the expected error $\mathbb{E}[\mu(E_1)\mathbb{1}[Y_j = 0] + \mu(E_0)\mathbb{1}[Y_j = 1]] = \frac{1}{4k_n}$. There are at least $k_n$ such intervals where $\mathcal{A}$ does not query on $[n - 1, n)$ which implies that the minimum expected error is equivalent to $\frac{1}{4}$.

Since we decomposed $\mathbb{R}_{\geq 0} = \bigcup_{n=1}^{\infty} [n - 1, n)$, then $\lim_{T \to \infty} \mathbb{E}\left[ \int_0^T \mathbb{1}[\mathcal{A}(X_t) \neq Y_t] \, dt \right] \geq \lim_{T \to \infty} \sum_{n=1}^{\lfloor T \rfloor} \mathbb{E}\left[ \int_{n-1}^n \mathbb{1}[\mathcal{A}(X_t) \neq Y_t] \, dt \right] \geq \lim_{T \to \infty} \sum_{n=1}^{\lfloor T \rfloor} \frac{1}{4} = \infty$. We used a randomized method to construct a family of continuous processes such that the expected error of a randomly chosen process reaches $\infty$. Applying the probabilistic method, there exists a continuous process whose expected error also reaches $\infty$. Therefore, $MB_{\mathcal{P}(H)}(\mathcal{A}) = \sup_{(Z_t)_{t \geq 0}} MB_{\mathcal{P}(H)}(\mathcal{A}, (Z_t)_{t \geq 0}) = \infty$ for any learning algorithm $\mathcal{A}$ with a linear querying strategy $Q_{\mathcal{A}}(t)$ so $H$ is unlearnable. $\square$

# B    Learnability of Pattern Classes from Discrete Data Streams

## B.1    Query-based Feedback Online Learning

In this section, we are interested in developing a theory of realizable learning of pattern classes under discrete data streams in the blind-prediction setting. While a general theory of pattern classes under discrete data streams would involve considering the agnostic case as well, we focus on the simplest scenario which is realizable learning under a binary label space $\mathcal{Y} = \{0, 1\}$ assuming deterministic learning algorithms. While this learning setting might seem quite restrictive, no such theory exists for the learnability of general pattern classes so we provide the first results in this space. We also develop this theory in the blind-prediction setting and a future direction of this work would be to characterize it in the update-and-deploy setting.

In the blind-prediction setting, assume a non-empty discrete pattern class $\mathcal{P}$ and some budget of queries $Q$. Assume a deterministic learning algorithm. One full round in this setting occurs in the following fashion at every $t \in \mathbb{N}$:

1. The learner makes a prediction $\hat{Y}_t \in \mathcal{Y}$ and decides to query or not.

2. The learner reveals $\hat{Y}_t$.

3. The adversary selects the true label $Y_t$.

4. If the learner does query, then the adversary reveals $(X_t, Y_t)$ to the learner.

The primary constraints governing this setting are realizability with respect to $\mathcal{P}$. Letting $(Z_t)_{t=1}^\infty = (X_t, Y_t)_{t=1}^\infty$ be the sequence of data points and true labels, then $(Z_t)_{t=1}^\infty \in \mathcal{P}$ to be considered realizable. It's important to note that the learner at any given time $t$ makes a prediction $\hat{Y}_t$ based solely on the current timestamp and history of previous queries.

## B.2 Query-based Mistake-Bounds

We turn to mistake-bounds to effectively capture the minimum number of mistakes an optimal deterministic learning algorithm will make.

The number of mistakes a deterministic learning algorithm $\mathcal{A}$ makes given a target pattern/discrete data stream $P^* = (Z_t)_{t=1}^\infty \in \mathcal{P}$ and a budget of $Q$ queries is denoted as $M_Q(\mathcal{A}, P^*)$. Formally, $M_Q(\mathcal{A}, P^*)$ can be understood as

$$M_Q(\mathcal{A}, P^*) = \sum_{i=1}^\infty \mathbb{1}[\mathcal{A}(X_t) \neq Y_t].$$

To consider the mistake-bound of the learning algorithm $\mathcal{A}$ on $\mathcal{P}$ given a budget of $Q$ queries, we get

$$M_Q(\mathcal{A}, \mathcal{P}) = \sup_{P^* \in \mathcal{P}} M_Q(\mathcal{A}, P^*).$$

Finally, to obtain the optimal mistake-bound on $\mathcal{P}$ given $Q$ queries, we take the infimum over all deterministic learning algorithms.

$$M_Q(\mathcal{P}) = \inf_{\mathcal{A}} M_Q(\mathcal{A}, \mathcal{P})$$

### B.2.1 Query Trees

In this section, we describe a certain type of tree, the *query tree*, which we will be used to capture the learning framework explained in Section B.1. Given the budget of queries $Q$ as an input, the query tree can be used to depict the evolution of the game in the blind-prediction setting. The name query tree comes from the fact that these trees describe the evolution of the game from the learner's perspective through queries. As a result, these trees are used in conjunction to provide upper and lower bounds on the optimal mistake-bounds for deterministic learning algorithms.

**Definition B.1** (Query Tree). *A query tree $T$ is defined as a tuple $(\mathcal{V}, E, Q)$ where $\mathcal{V}$ is a collection of nodes, $E$ is a collection of edges, and $Q$ is the query budget.*

- *$T$ is a rooted binary tree where each node has at most two children which are referred to as the left child and the right child.*

- *Each node $V_j \in \mathcal{V}$ corresponds to some timestamp $t_i$ where $V_j^t = t_i$.*

- *The root node is represented by $V_1 \in \mathcal{V}$ and corresponds to $V_1^t = t_1$ where $t_1$ is the timestamp of the first query.*

- *$\forall V \in \mathcal{V}$, if $V' = \mathrm{Parent}(V)$, then $V'^t < V^t$.*

- *$\forall e \in E$ where $e = (V', V)$ with $V' = \mathrm{Parent}(V)$, then the edge weight is defined as $\omega(e) = y$ with $y = 0$ if $V = \mathrm{LeftChild}(V')$ or $y = 1$ if $V = \mathrm{RightChild}(V')$.*

- *Every root-to-leaf path $(V_1, ..., V_n)$ has $n = Q + 1$ nodes.*

### B.2.2 Query Learning Distance - Blind-Prediction Setting

In this section, we describe a dimension based on a family of query trees that correctly characterizes the complexity of learning pattern classes in the blind-prediction setting. In the following subsections, we first tackle the case when $Q = 0$ and then characterize the general setting for $Q > 0$.

**Special Variant: Blind Learning**   What if the learner wasn't allowed to even query once? How would this affect the number of mistakes the learner would make? We call this special setting the blind learning scenario since the learner receives absolutely no feedback on any round of the game; only the current time-step is given as input. While Section B.1 describes the exact procedure round-by-round, a closer look at the intricacies of this scenario can reduce the game into a simple two-step procedure. The learner is not allowed to query a single time; so this implies that the learner cannot use information about the sequence itself to update its algorithm. Additionally, since the learner is a deterministic learning algorithm, this implies that an all-knowing adversary has complete knowledge about the learner's prediction at every timestamp. Combining these two facts together, it follows that the learner's predictions are independent of the true labels and the adversary can simply select the sequence of true labels all at once. Formally speaking, the entire game can be described in the following two steps:

1. The learner selects a prediction vector $\hat{\mathbf{y}} \in \{0,1\}^\infty$.

2. The adversary selects the true outcome vector $\mathbf{y} \in \{0,1\}^\infty$.

Then, the number of mistakes is equivalent to $\sum_{i=1}^\infty \mathbb{1}[\hat{\mathbf{y}}(i) \neq \mathbf{y}(i)] = |\hat{\mathbf{y}} - \mathbf{y}|$ where $|\cdot|$ stands for the L1-norm. As is consistent with the framework described in Section B.1, the vector $\mathbf{y}$ must be realizable with respect to the pattern class $\mathcal{P}$ such that $\exists P \in \mathcal{P}$ where $\forall (X_t, Y_t) \in P, Y_t = \mathbf{y}(t)$. Below, we present an important lemma that characterizes the optimal mistake-bound $M_0(\mathcal{P})$.

**Lemma B.2.** *If the number of queries $Q = 0$, then*

$$M_0(\mathcal{P}) = \text{BlindLearningDimension}(\mathcal{P}) = \inf_{\hat{\mathbf{y}} \in \{0,1\}^\infty} \sup_{\mathbf{y} \in \mathcal{P}^y} |\hat{\mathbf{y}} - \mathbf{y}|$$

*where $|\cdot|$ represents the L1-distance and $\mathcal{P}^y = \{\mathbf{y} \in \{0,1\}^\infty : \exists P \in \mathcal{P} \text{ s.t. } \forall (X_t, Y_t) \in P, Y_t = \mathbf{y}(t)\}$ which represents the set of infinite binary vectors that are realizable with respect to $\mathcal{P}$.*

*Proof.* We divide this proof into two parts by devoting the first half to a lower bound proof showing that $\text{BlindLearningDimension}(\mathcal{P}) \leq M_0(\mathcal{P})$. The second half of the proof is devoted to showing that there exists an algorithm $\mathcal{A}$ such that $M_0(\mathcal{A}, \mathcal{P}) = \text{BlindLearningDimension}(\mathcal{P})$. Finally, we combine these two statements to ultimately show that $\text{BlindLearningDimension}(\mathcal{P}) = M_0(\mathcal{P})$.

We first show the lower-bound proof by letting $\mathcal{A}$ be any deterministic learning algorithm. Then, $M_0(\mathcal{A}, P)$ is the mistake-bound of the learner $\mathcal{A}$ given the pattern $P \in \mathcal{P}$. Let $\mathbf{y}'$ be the output of $\mathcal{A}$ given no queries. This is equivalent to $M_0(\mathcal{A}, P) = |\mathbf{y}' - \mathbf{y}|$ where $(\mathbf{x}, \mathbf{y}) = P$. $M_0(\mathcal{A}, \mathcal{P})$ is the maximum mistake-bound of the learning algorithm $\mathcal{A}$ over the entire pattern class $\mathcal{P}$. More technically, we represent $M_0(\mathcal{A}, \mathcal{P}) = \sup_{\mathbf{y} \in \mathcal{P}^y} |\mathbf{y}' - \mathbf{y}|$. Since $\mathbf{y}' \in \{0,1\}^\infty$, it then holds that $\inf_{\hat{\mathbf{y}} \in \{0,1\}^\infty} \sup_{\mathbf{y} \in \mathcal{P}^y} |\hat{\mathbf{y}} - \mathbf{y}| \leq \sup_{\mathbf{y} \in \mathcal{P}^y} |\mathbf{y}' - \mathbf{y}|$. It follows that $\text{BlindLearningDimension}(\mathcal{P}) \leq M_0(\mathcal{A}, \mathcal{P})$. Since $\mathcal{A}$ was an arbitrary deterministic learning algorithm, it then follows that $\text{BlindLearningDimension}(\mathcal{P}) \leq M_0(\mathcal{P})$.

For the upper bound proof we focus on the case when $\text{BlindLearningDimension}(\mathcal{P}) < \infty$ since $M_0(\mathcal{P}) = \infty$ when $\text{BlindLearningDimension}(\mathcal{P}) = \infty$. Let $\mathcal{A}$ be a deterministic learning algorithm that predicts the vector $\hat{\mathbf{y}}$ such that $\sup_{\mathbf{y} \in \mathcal{P}^y} |\hat{\mathbf{y}} - \mathbf{y}| = \inf_{\hat{\mathbf{y}} \in \{0,1\}^\infty} \sup_{\mathbf{y} \in \mathcal{P}^y} |\hat{\mathbf{y}} - \mathbf{y}|$. Since $\text{BlindLearningDimension}(\mathcal{P}) < \infty$ and $\mathbf{y}$ is a binary vector, then there exists a vector $\hat{\mathbf{y}}$ achieving the minimum. It directly follows that $M_0(\mathcal{A}, \mathcal{P}) = \text{BlindLearningDimension}(\mathcal{P})$. Since $M_0(\mathcal{P}) \leq M_0(\mathcal{A}, \mathcal{P})$, then $M_0(\mathcal{P}) \leq \text{BlindLearningDimension}(\mathcal{P})$. By combining the lower bound and upper bound statements, we get the following inequality $\text{BlindLearningDimension}(\mathcal{P}) \leq M_0(\mathcal{P}) \leq \text{BlindLearningDimension}(\mathcal{P})$ so $M_0(\mathcal{P}) = \text{BlindLearningDimension}(\mathcal{P})$.

$\square$

**General Setting**   We now define the dimension $QLD$ or query learning distance on these family of query trees $\mathcal{T}$ realizable with respect to the discrete pattern class $\mathcal{P}$ given $Q$ queries. The $QLD$ quantity can be thought as analogous to the notion of rank of a binary tree but setup in a slightly different fashion. Below, for each $T \in \mathcal{T}$, we describe the query learning distance.

$$QLD_T(\mathcal{P}, Q, i) = \mathbb{1}[i = Q] \cdot \text{BlindLearningDimension}(\mathcal{P}) + \mathbb{1}[i < Q] \left( \inf_{\hat{\mathbf{y}} \in \{0,1\}^{t_i - (t_{i-1}+1)}} \right.$$

$$\left. \sup_{\substack{x_{t_i} \in \mathcal{X} \\ \mathbf{y} \in \{0,1\}^{t_i - (t_{i-1}+1)}}} |\hat{\mathbf{y}} - \mathbf{y}| \cdot \mathbb{1}[\mathcal{P}_{(\star, \mathbf{y})} \neq \emptyset] + \begin{cases} \max\{T_0, T_1\} & \text{if } T_0 \neq T_1 \\ T_0 + 1 & \text{else} \end{cases} \right) \quad (1)$$

$$\text{where} \qquad T_0 = QLD_{T_L}(\mathcal{P}_{(\star, \mathbf{y})(x_{t_i}, 0)}, Q, i+1) \tag{2}$$

$$T_1 = QLD_{T_R}(\mathcal{P}_{(\star, \mathbf{y})(x_{t_i}, 1)}, Q, i+1). \tag{3}$$

In Eqs. 1, 2, and 3, $T_L$ is the left subtree of $T$, $T_R$ is the right subtree of $T$, $\hat{\mathbf{y}}$ is the sequence of predictions, $\mathbf{y}$ is the sequence of true labels, $x_{t_i}$ is the instance at time $t_i$, and $\mathcal{P}_{(\star, \mathbf{y})} = \{P \in \mathcal{P} : \forall y_t \in \mathbf{y}, P^y(t) = y_t\}$ with $P^y(t)$ referring to the $t^{th}$ label of pattern $P$. Additionally, $t_i$ and $t_{i-1}$ refer to the timestamps of the $i^{th}$ and $i-1^{th}$ queries respectively that correspond to the root-to-leaf path dictated by the recursion.

**Defining QLD$(\mathcal{P}, \mathbf{Q})$**   Let $\mathcal{T}(\mathcal{P}, Q)$ be the collection of all query trees for the predict-then-query setting that are realizable with respect to $\mathcal{P}$ and contains $Q$ query nodes on each branch. Then, $\mathcal{T}^k(\mathcal{P}, Q) = \{T \in \mathcal{T}(\mathcal{P}, Q) : QLD_T(\mathcal{P}, Q, 0) = k\}$ is the collection of trees whose query learning distance is exactly $k$. Finally, we define $QLD(\mathcal{P}, Q) = \inf\{k \in \mathbb{N} \cup \{0\} : \mathcal{T}^k(\mathcal{P}, Q) \neq \emptyset\}$.

**Lemma B.3.** *For any discrete pattern class $\mathcal{P}$ and query budget $Q \in \mathbb{N} \cup \{0\}$, $QLD(\mathcal{P}, Q) \leq M_Q(\mathcal{P})$.*

*Proof.* Let $\mathcal{A}$ be any deterministic learning algorithm. A proof by induction will be established on the pair $(\mathcal{P}, Q)$ taking $Q = 0$ to be the base. We refer to Lemma B.2 to show that $\forall \mathcal{P}' \subseteq \mathcal{P}, QLD(\mathcal{P}', 0) = M_0(\mathcal{P}') \leq M_0(\mathcal{A}, \mathcal{P}')$. Now, we apply the inductive step on $(\mathcal{P}', Q')$ where $\mathcal{P}' \subseteq \mathcal{P}$ and $Q' < Q$, then $QLD(\mathcal{P}', Q') \leq M_{Q'}(\mathcal{A}, \mathcal{P}')$. The rest of the proof is devoted to showing that $QLD(\mathcal{P}, Q) \leq M_Q(\mathcal{A}, \mathcal{P})$ by describing an adversarial strategy that guarantees this bound.

Let $t_1 \in \mathbb{N}$ be the first query timestamp made by the learning algorithm $\mathcal{A}$. Since $\mathcal{A}$ is a deterministic learner, the adversary has knowledge of $t_1$. To narrow down its selection of the true labels for the first $t_1$ rounds, the adversary can select an optimal query tree $T$ based on the value $QLD_T(\mathcal{P}, Q, 0)$ given that the root node has $V_1^t = t_1$. Given that $QLD_T(\mathcal{P}, Q, 0)$ follows the piece-wise function described in Eq. 1, the adversary can select the larger of $T_0$ or $T_1$ (if equal, $T_0$ is chosen). Without loss of generality, let $T_0$ be the subtree chosen by the adversary. For the first $t_1 - 1$ rounds, let the adversary selects the optimal vector of true labels $\mathbf{y}$ given knowledge of the procedure of algorithm $\mathcal{A}$. Let $\hat{\mathbf{y}}$ represent the vector of predicted labels by the learner for the first $t_1 - 1$ rounds. At time $t_1$, the learner will present its prediction $\hat{y}_{t_1}$. The adversary can select $x_{t_1}$ that corresponds to the supremum in Eq. 1 and set the true label $y_{t_1} = 0$. In the special case that $T_0 = T_1$, then $y_{t_1} = 1 - \hat{y}_{t_1}$.

Then the number of mistakes made by the learner in the first $t_1$ rounds is equivalent to $|\hat{\mathbf{y}} - \mathbf{y}| + \mathbb{1}[y_{t_1} \neq \hat{y}_{t_1}]$. On the remaining number of rounds, $M_{Q-1}(\mathcal{A}, \mathcal{P}_{(\star, \mathbf{y})(x_{t_1}, y_{t_1})})$ represents the optimal mistake-bound of the learner $\mathcal{A}$. Using the inductive step, we can show that $|\hat{\mathbf{y}} - \mathbf{y}| + \mathbb{1}[y_{t_1} \neq \hat{y}_{t_1}] + QLD(\mathcal{P}_{(\star, \mathbf{y})(x_{t_1}, y_{t_1})}, Q - 1) \leq |\hat{\mathbf{y}} - \mathbf{y}| + \mathbb{1}[y_{t_1} \neq \hat{y}_{t_1}] + M_{Q-1}(\mathcal{A}, \mathcal{P}_{(\star, \mathbf{y})(x_{t_1}, y_{t_1})})$. Since $M_Q(\mathcal{A}, \mathcal{P})$ is calculated as the supremum over all adversarial approaches given the algorithm $\mathcal{A}$, then $M_Q(\mathcal{A}, \mathcal{P}) \geq |\hat{\mathbf{y}} - \mathbf{y}| + \mathbb{1}[y_{t_1} \neq \hat{y}_{t_1}] + M_{Q-1}(\mathcal{A}, \mathcal{P}_{(\star, \mathbf{y})(y_{t_1}, y_{t_1})})$. Now, we show that $QLD(\mathcal{P}, Q) \leq |\hat{\mathbf{y}} - \mathbf{y}| + \mathbb{1}[y_{t_1} \neq \hat{y}_{t_1}] + QLD(\mathcal{P}_{(\star, \mathbf{y})(x_{t_1}, y_{t_1})}, Q - 1)$. Assume that the predictions made by algorithm $\mathcal{A}$ induce $|\hat{\mathbf{y}} - \mathbf{y}| + \mathbb{1}[y_{t_1} \neq \hat{y}_{t_1}] + QLD(\mathcal{P}_{(\star, \mathbf{y})(x_{t_1}, y_{t_1})}, Q - 1) < QLD(\mathcal{P}, Q)$. Since the adversary's selection of true labels is an optimal label vector given the workings of learning algorithm $\mathcal{A}$, then the adversary's decision aligns with the supremum in Eq. 1. Then, there must exist a tree $T'$ whose largest distance is equal to that value with $t_1$ being the timestamp of the root node. Formally speaking, this implies the existence of $T'$ such that $QLD_{T'}(\mathcal{P}, Q, 0) < QLD(\mathcal{P}, Q)$. If such a tree existed, then the adversary would have selected $T'$ which violates the minimality of $QLD(\mathcal{P}, Q)$ and the assumption that the adversary chose the most minimal tree satisfying $V_1^t = t_1$. As a result, $QLD(\mathcal{P}, Q) \leq |\hat{\mathbf{y}} - \mathbf{y}| + \mathbb{1}[y_{t_1} \neq \hat{y}_{t_1}] + QLD(\mathcal{P}_{(\star, \mathbf{y})(x_{t_1}, y_{t_1})}, Q - 1)$. Placing all the inequalities

together, we get $QLD(\mathcal{P}, Q) \leq |\hat{\mathbf{y}} - \mathbf{y}| + \mathbb{1}[y_{t_1} \neq \hat{y}_{t_1}] + QLD(\mathcal{P}_{(\star,\mathbf{y})(x_{t_1},y_{t_1})}, Q-1) \leq |\hat{\mathbf{y}} - \mathbf{y}| + \mathbb{1}[y_{t_1} \neq \hat{y}_{t_1}] + M_{Q-1}(\mathcal{A}, \mathcal{P}_{(\star,\mathbf{y})(x_{t_1},y_{t_1})}) \leq M_Q(\mathcal{A}, \mathcal{P})$ which results in $QLD(\mathcal{P}, Q) \leq M_Q(\mathcal{A}, \mathcal{P})$. Since $\mathcal{A}$ was an arbitrary learning algorithm, then it holds that $QLD(\mathcal{P}, Q) \leq M_Q(\mathcal{P})$. $\qquad \square$

In Algorithm 3 detailed below, we denote by $\mathbf{y}_1 \circ \mathbf{y}_2$ the vector obtained by concatenating vector $\mathbf{y}_2$ after vector $\mathbf{y}_1$. Additionally, the usage of $T_0$ and $T_1$ refer to Eqs. 2 and 3 respectively. If $T_{y_t}$ is used, this implies a query subtree that was either the left child if $y_t = 0$ or the right child if $y_t = 1$.

---

**Algorithm 3** BP-SOA$(\mathcal{P}, Q)$

---

**Require:** $\mathcal{P} \neq \emptyset$
**Require:** $Q \geq 0$
1: $\hat{\mathbf{x}}_h$ = history of previously observed instances
2: $\hat{\mathbf{y}}_l$ = list of current predictions
3: $O$ = history of previous queries {Elements of $O$ are $(t, y)$ where $t$ is time, $y$ is label}
4: $i = 0, t_i = 1$ {Initial query number and timestamp}
5: Select tree $T$ such that $QLD(\mathcal{P}, Q, 0) = QLD(\mathcal{P}, Q)$, $t_i = V_1^t$
6: $T_{\text{end}} = \infty$
7: **for** $t = 1$ to $T_{\text{end}}$ **do**
8:   **if** $t < t_i$ **then**
9:     **if** $t = 1$ **then**
10:

$$\hat{\mathbf{y}} = \arg\min_{\hat{\mathbf{y}} \in \{0,1\}^{t_i-1}} \sup_{\mathbf{y} \in \{0,1\}^{t_i-1}} |\hat{\mathbf{y}} - \mathbf{y}| \cdot \mathbb{1}[\mathcal{P}_{(\star,\mathbf{y})} \neq \emptyset] + QLD(\mathcal{P}_{(\star,\mathbf{y})}, Q)$$

11:       Append $\hat{\mathbf{y}}$ to $\hat{\mathbf{y}}_l$
12:     **end if**
13:     Predict $\hat{\mathbf{y}}_l(t)$, add $\star$ to $\hat{\mathbf{x}}_h$
14:   **else if** $i < Q$ **then**
15:     $\hat{y}_t = \arg\max_{r \in \{0,1\}} \sup_{x_{t_i} \in \mathcal{X}} \sup_{\substack{\mathbf{y} \in \{0,1\}^{t_i-1} \\ \forall(t,y) \in O, \mathbf{y}(t)=y}} |\hat{\mathbf{y}}_l - \mathbf{y}| \mathbb{1}[\mathcal{P}_{(\hat{\mathbf{x}}_h,\mathbf{y})} \neq \emptyset] + $

    $QLD_{T_r}(\mathcal{P}_{(\hat{\mathbf{x}}_h,\mathbf{y})(x_{t_i},r)}, Q, i+1)$
16:     Predict $\hat{y}_t$ and add $\hat{y}_t$ to $\hat{\mathbf{y}}_h$
17:     Receive $(x_t, y_t)$, add $(t, y_t)$ to $O$, and add $x_t$ to $\hat{\mathbf{x}}_h$
18:     Set $i = i + 1$, $t_i = V_1^t$ {$V_1$ is the root node of $T_{y_t}$}
19:

$$\hat{\mathbf{y}} = \arg\min_{\hat{\mathbf{y}} \in \{0,1\}^{t_i-(t_{i-1}+1)}} \sup_{x_{t_i} \in \mathcal{X}} \sup_{\substack{\mathbf{y} \in \{0,1\}^{t_i-1} \\ \forall(t,y) \in O, \mathbf{y}(t)=y}} |\hat{\mathbf{y}}_l \circ \hat{\mathbf{y}} - \mathbf{y}| \cdot \mathbb{1}[\mathcal{P}_{(\hat{\mathbf{x}}_h \circ \{\star\}, \mathbf{y})} \neq \emptyset] + \begin{cases} \max\{T_0, T_1\} & \text{if } T_0 \neq T_1 \\ T_0 + 1 & \text{else} \end{cases}$$

20:     Append $\hat{\mathbf{y}}$ to $\hat{\mathbf{y}}_l$
21:   **else**
22:     $\mathbf{y}' = \arg\max_{\substack{\mathbf{y} \in \{0,1\}^{t_i} \\ \forall(t,y) \in O, \mathbf{y}(t)=y}} |\hat{\mathbf{y}} - \mathbf{y}| \cdot \mathbb{1}[\mathcal{P}_{(\hat{\mathbf{x}}_h,\mathbf{y})} \neq \emptyset] + \text{BlindLearningDimension}(\mathcal{P}_{(\hat{\mathbf{x}}_h,\mathbf{y})})$
23:     Let $\hat{\mathbf{y}}$ be such that $\sup_{\mathbf{y} \in \mathcal{P}^y_{(\hat{\mathbf{x}}_h,\mathbf{y}')}} |\hat{\mathbf{y}} - \mathbf{y}| = \inf_{\hat{\mathbf{y}} \in \{0,1\}^\infty} \sup_{\mathbf{y} \in \mathcal{P}^y_{(\hat{\mathbf{x}}_h,\mathbf{y}')}} |\hat{\mathbf{y}} - \mathbf{y}|$
24:     Append $\hat{\mathbf{y}}$ to $\hat{\mathbf{y}}_l$
25:     Set $T_{\text{end}} = t_i$
26:   **end if**
27: **end for**
28: **for** $t = t_i$ to $\infty$ **do**
29:   Predict $\hat{\mathbf{y}}_l(t)$
30: **end for**

---

**Lemma B.4.** *For any discrete pattern class $\mathcal{P}$ and query budget $Q \in \mathbb{N} \cup \{0\}$, $M_Q(\mathcal{P}) \leq QLD(\mathcal{P}, Q)$.*

*Proof.* Let $\mathcal{A}$ be the BP-SOA which is detailed in Algorithm 3. A proof by induction will be established on the pair $(\mathcal{P}, Q)$ taking $Q = 0$ to be the base case. Let $\mathcal{P}' \subseteq \mathcal{P}$. In the base case, we execute Algorithm 3 with the inputs $\mathcal{P}'$ and $Q = 0$. Since $Q = 0$, Algorithm 3 selects the vector $\hat{\mathbf{y}}$ corresponding to the BlindLearningDimension$(\mathcal{P}')$ and appends it to $\hat{\mathbf{y}}_l$ (lines 22-24). Then, $\mathcal{A}$ skips to lines 28-30 making predictions according to $\hat{\mathbf{y}}_l$. Then, we refer to Lemma B.2 to show that BlindLearningDimension$(\mathcal{P}') \leq M_0(\mathcal{P}') \leq M_0(\text{Algorithm 3}, \mathcal{P}') \leq$ BlindLearningDimension$(\mathcal{P}')$ implying that $M_0(\mathcal{P}') = \text{BlindLearningDimension}(\mathcal{P}')$.

Now, we apply the inductive step on $(\mathcal{P}', Q')$ where $\mathcal{P}' \subseteq \mathcal{P}$ and $Q' < Q$, then $M_{Q'}(\mathcal{A}, \mathcal{P}') \leq QLD(\mathcal{P}', Q')$. The rest of the proof is devoted to showing that $M_Q(\mathcal{A}, \mathcal{P}) \leq QLD(\mathcal{P}, Q)$.

From Algorithm 3, we know that $\mathcal{A}$ selects the query tree $T$ such that $QLD_T(\mathcal{P}, Q, 0) = QLD(\mathcal{P}, Q)$ on line 5 with the first query timestamp $t_1 = V_1^t$. For rounds $t < t_1$ rounds, Algorithm 3 will select its predictions $\hat{y}_t$ that minimizes the optimization expression in lines 10 and 19 based on the history of previous queries. On round $t_1$, Algorithm 3 selects $\hat{y}_{t_1}$ in line 15 based on the larger subtree, $T_0$ or $T_1$. Without loss of generality, assume that $T_0$ is the larger subtree and in the case of a tie, $T_0$ is selected. Then, $\hat{y}_{t_1} = 0$ and the learner receives $(x_{t_1}, y_{t_1})$ after querying.

It follows that the mistakes made by the learner on the first $t_1$ rounds correspond to $|\mathbf{y} - \hat{\mathbf{y}}| + \mathbb{1}[\hat{y}_{t_1} \neq y_{t_1}]$ where $\mathbf{y}$ and $y_{t_1}$ represent the true labels selected by the adversary. Since the adversary is operating under the constraint of realizability, then it must hold that $\mathcal{P}_{(\star, \mathbf{y})(x_{t_1}, y_{t_1})} \neq \emptyset$ where $\star$ is a placeholder for any sequence of instances satisfying the constraint. From the inductive step, it follows that $|\hat{\mathbf{y}} - \mathbf{y}| + \mathbb{1}[\hat{y}_{t_1} \neq y_{t_1}] + M_{Q-1}(\mathcal{A}, \mathcal{P}_{(\star, \mathbf{y})(x_{t_1}, y_{t_1})}) \leq |\hat{\mathbf{y}} - \mathbf{y}| + \mathbb{1}[\hat{y}_{t_1} \neq y_{t_1}] + QLD(\mathcal{P}_{(\star, \mathbf{y})(x_{t_1}, y_{t_1})}, Q - 1)$. Assume that the adversary's choices of instances and true labels on the first $t_1$ rounds yield $|\hat{\mathbf{y}} - \mathbf{y}| + \mathbb{1}[\hat{y}_{t_1} \neq y_{t_1}] + QLD(\mathcal{P}_{(\star, \mathbf{y})(x_{t_1}, y_{t_1})}, Q - 1) > QLD(\mathcal{P}, Q)$. Since $\mathcal{A}$ selected tree $T$, this implies that $QLD_T(\mathcal{P}, Q, 0) = QLD(\mathcal{P}, Q)$. Additionally, $\mathcal{A}$ always selects the predictions that minimizes over the worst possible game outcomes (line 19 of Algorithm 3) with the query prediction aligning with that of the larger subtree. As a result $|\hat{\mathbf{y}} - \mathbf{y}| + \mathbb{1}[\hat{y}_{t_1} \neq y_{t_1}] + QLD_{T_L}(\mathcal{P}_{(\star, \mathbf{y})(x_{t_1}, y_{t_1})}, Q, 1) \leq QLD(\mathcal{P}, Q)$. And by definition $QLD(\mathcal{P}_{(\star, \mathbf{y})(x_{t_1}, y_{t_1})}, Q - 1) \leq QLD_{T_L}(\mathcal{P}_{(\star, \mathbf{y})(x_{t_1}, y_{t_1})}, Q, 1)$, so it must hold that $|\hat{\mathbf{y}} - \mathbf{y}| + \mathbb{1}[\hat{y}_{t_1} \neq y_{t_1}] + QLD(\mathcal{P}_{(\star, \mathbf{y})(x_{t_1}, y_{t_1})}, Q - 1) \leq QLD(\mathcal{P}, Q)$. As a result, $|\hat{\mathbf{y}} - \mathbf{y}| + \mathbb{1}[\hat{y}_{t_1} \neq y_{t_1}] + M_{Q-1}(\mathcal{A}, \mathcal{P}_{(\star, \mathbf{y})(x_{t_1}, y_{t_1})}) \leq QLD(\mathcal{P}, Q)$. Since this inequality holds for any choice of $\mathbf{y}$, $x_{t_1}$, and $y_{t_1}$, it follows that $M_Q(\mathcal{A}, \mathcal{P}) \leq QLD(\mathcal{P}, Q)$. Since $M_Q(\mathcal{P}) \leq M_Q(\mathcal{A}, \mathcal{P})$, we show that $M_Q(\mathcal{P}) \leq QLD(\mathcal{P}, Q)$. $\square$

