# OpenReview forum: "Learning from Snapshots of Discrete and Continuous Data Streams"
_NeurIPS.cc/2024/Conference — NeurIPS 2024 poster_

### Official Review · Reviewer_cGKc · 2024-07-11

**Soundness:** 4
**Presentation:** 3
**Contribution:** 4
**Rating:** 6
**Confidence:** 2

**Summary:**

This paper considers the problem of online learning in the setting of discrete and continous data streams. It first introduces two novel learning frameworks: the update-and-deploy setting and the blind-prediction setting. The update-and-deploy setting allows a learning algorithm to discretely query a data stream to update a predictor, while the blind-prediction setting involves making predictions without observing the current data value but only based on the previous querious and current timestamp. The authors proves the error bound when a non-adaptive learner uniformly samples queries in the update-and-deploy setting. Then in the blind-prediction setting, they show that no matter the learning algorithm is adaptive or non-adaptive, it is not learnable for any concept class. It also discusses the characterization of learning algorithms should have when learning pattern classes in the continous data stream. Finally, it extends it results to the discrete data streams for the second framwork for deterministic algorithms.

**Strengths:**

1. It proposes two novel framework for learning under continous data streams and provide substantial theoretical analysis to show the error bound for the first setting and the findings in the second setting.
2. It has substantial contributions regarding the novel frameworks, discussion about the different learners to learn the pattern class, and extends the results into discrete data stream.
3. It proposes the new mesaure called the query-learning distance(QLD) and the algorithm they develop has an optimal mistake-bound with respect to the new QLD.

**Weaknesses:**

1. The paper lacks empirical validation of the results, and there's no comparison with other existing possible solutions or natural extensions of existing solutions to their framwork.

**Questions:**

Is it trivial to consider what kind of non-adaptive learners should be when learning pattern classes under continous data streams?

**Limitations:**

see weakness.

---

> ### Author Rebuttal · Authors · 2024-08-07
>
> Thank you for your positive comments towards the contributions made by our two learning frameworks and algorithms!
>
> ### Weaknesses
> **Comment 1**: This paper is primarily a learning-theoretic paper so it's focused on establishing a concrete theory in terms of mathematical statements and proofs. Since the main purpose of our paper is to develop a theory within online learning, we would defer empirical evaluation to future application-oriented research.
>
> ### Questions:
> **Question 1**: This is an important question regarding the learnability of pattern classes under a continuous data stream. When we consider learnability of some pattern class $\mathcal{P}$, this means that we look at all possible learning algorithms and if there exists a learning algorithm whose mistake-bound is finite on $\mathcal{P}$, then we consider $\mathcal{P}$ to be learnable. The pattern class example in Section 4.2 illuminates an important pattern class that is only learnable under an adaptive learning algorithm. Since we have shown that there exists pattern classes that are not learnable by non-adaptive learners but learnable by an adaptive algorithm, this means that the criteria for learnability of pattern classes is beyond the scope of non-adaptive learners.

---

> > ### Comment · Reviewer_cGKc · 2024-08-11
> >
> > Thanks for the author's feedback on the weaknesses and questions. I do feel that the technical contributions in this paper are solid but adding experiments would definitely enhance and make the algorithms more concrete. I intend to keep my score, good luck with the submission.

---

### Official Review · Reviewer_dysh · 2024-07-12

**Soundness:** 3
**Presentation:** 2
**Contribution:** 3
**Rating:** 6
**Confidence:** 2

**Summary:**

The paper studies the online learning problem where the algorithms are receiving a stream of $(X_i, Y_i)$ in which $X_i$ is an instance and $Y_i$ is the corresponding label, and in each time point  $i$, the algorithm is required to make a prediction $\hat{Y}_i$, which is based on the historical data and the current instance (or not). The paper studies the problem in several settings:

For continuous case:

1. In the update-and-deploy setting, the paper proposes a non-adaptive algorithm that achieves a mistake bound $LD(H)$, where the main idea of the learner is using sampling.

2. In the blinding prediction setting,  the paper shows that any learning algorithm, adaptive or non-adaptive, is not learnable in the blind-prediction setting.

3. The paper considers the case where the algorithms are required to learn pattern classes. Then it designs a continuous pattern class that any random sampling algorithms fail but there is an adaptive learning algorithm that successfully learns P with zero expected error.  This shows a separation of the adaptive and non-adaptive case in this setting.

Finally, the paper develops a theory for the discrete case where they characterize a combinatorial quantity QLD where there is a deterministic learning algorithm whose optimal mistake-bound is equivalent to it.

**Strengths:**

1. This is a strong theory paper. To get this results, the paper considers several algorithmic ideas which are interesting to me (though I am not an expert in the area of online learning).

**Weaknesses:**

1. The writing of this paper is not good enough. When I first read this paper, I felt like some parts of the paper were not easy to follow and need to be improved.  For example:

-- In Section 1.3, several notations are not defined until the later Section.

-- The definition of the littlestone dimension is not mentioned.

**Questions:**

See the above question.

---

> ### Author Rebuttal · Authors · 2024-08-07
>
> Thank you for your comments regarding the strength of our paper!
>
> ### Weaknesses:
> **Comment 1**: Thank you for this comment. After a revisit to Section 1.3, many of the terms such as $MB_{\mathcal{P}(H)}$ and $LD(H)$ were not properly defined. We will fix this revision to make sure that the terms are defined before their usage in addition to their formal definitions in the later section.
>
> **Comment 2**: Thank you for this comment. We will include a brief section in the paper explaining the Littlestone dimension.

---

### Official Review · Reviewer_jfDD · 2024-07-14

**Soundness:** 3
**Presentation:** 3
**Contribution:** 4
**Rating:** 7
**Confidence:** 3

**Summary:**

This paper studies mistake bounds for discrete and continuous labelled data streams in two different coupling settings between labeller and learner (update-and-deploy and blind-prediction)

**Strengths:**

The paper is a theory paper that characterizes the learnability of pattern classes. The proof in the main paper is very accessible, easy-to-follow and accessible. I commend the authors for the accessibility of the proof in the main paper.

**Weaknesses:**

The paper feels overloaded with results for a 9-page NeurIPS submission. All the results for the adaptive learner case feel "crammed in" and, unless the reader is prepared to read the appendix, cannot be easily understood. I wish the authors would have constrained on the study of Algorithm 1; while less results, it would have made the paper into a self-contained theory paper with a significant result that may have had a proper conclusion.

**Questions:**

* in line 148, 166, 199 and 239, shouldn't it be $\in O(t)$ rather than $= O(t)$?
* the reader would greatly benefit from introducing the Littlestone dimension (LD)
* in line 267, shouldn't it say "expected size $\Delta$"?
* line 320: "are needed to" should read "are needed for"

**Limitations:**

The result is not allowing to discover new algorithms for this setting; it explains why a random query strategy works.

---

> ### Author Rebuttal · Authors · 2024-08-07
>
> Thank you for your comments about our paper's proofs! We are glad that it was highly accessible!
>
> ### Weaknesses:
> **Comment 1**: Thank you for this feedback. We agree that some structural changes can be made to the paper to make it more readable. The main highlights of this paper are to show that non-adaptive learning algorithms are powerful enough to learn concept classes from a continuous stream of data in the update-and-deploy setting and pattern classes require adaptive learning algorithms in either learning framework. One potential revision is to present these two results first with their full proofs so the reader doesn't have to reference the Appendix and understand what the main ideas are of the paper.
>
> ### Questions:
> **Question 1**: When describing the growth rate of a function in terms of Big-O notation, both $Q_{\mathcal{A}}(t) = O(t)$ and $Q_{\mathcal{A}}(t) \in O(t)$ convey the same meaning. The expression $Q_{\mathcal{A}}(t) = O(t)$ is considered an ``abuse of notation" but it implies that $Q_{\mathcal{A}}(t)$ is of order $O(t)$.
>
> **Question 2**: Thank you for this comment. We agree and we will add a brief section in the paper introducing the concept.
>
> **Question 3**: According to line $8$ in the pseudo-code of Algorithm $1$, the next query time $t_q$ is sampled in the following way: $t_q \sim \mathrm{Unif}[t, t + \Delta]$. The query time $t_q$ is selected from a uniform distribution over intervals of size $\Delta$ so it's not an expected size of $\Delta$ but rather consecutive query times are spaced out on average of size $\frac{\Delta}{2}$ since the mean of a uniform distribution on an interval of size $\Delta$ is $\Delta/2$.
>
> **Question 4**: Thank you for this edit.

---

> > ### Comment · Reviewer_jfDD · 2024-08-12
> >
> > Thanks for the detailed comments of the authors. I would still suggest to use $\in O(t)$ but otherwise I am fine with the answers and stick to my assessment. Good luck with the paper and hopefully see it at NeurIPS this year!

---

### Official Review · Reviewer_F4wW · 2024-07-16

**Soundness:** 2
**Presentation:** 2
**Contribution:** 3
**Rating:** 4
**Confidence:** 4

**Summary:**

This paper introduces a novel learning-theoretic framework for understanding online learning from continuous and discrete data streams through selective querying. The authors propose two settings: the update-and-deploy setting, where a learner updates a predictor based on queried data, and the blind-prediction setting, where predictions are made independently of the current data stream. They analyze the learnability of concept classes and pattern classes under these settings, providing theoretical bounds and algorithms.

**Strengths:**

- The framework presented in this paper for studying continuous data streams is interesting and novel in my opinion.
- The authors present simple algorithms which are interesting and derive results in terms of already existing concepts like Littlestone dimension which is interesting.

**Weaknesses:**

- There are certain parts in the paper that are unclear as described below. It would be nice if the examples presented in the paper to describe the settings and the theoretical framework are presented together for better understanding. It would be nice if the authors start from the traditional setting first and then add the elements relating to the continuous setting to clearly differentiate between the two.

**Questions:**

- The Update and Deploy setting and the Blind Prediction settings are not clear to me. In particular, we can think of the update in the Update and Deploy (update from the cloud translator) setting as a kind of feedback (hyper spectral image) in the Blind Prediction setting? In the formal definitions in sections 2.2 and 2.3, is the difference between the two settings is that in one f^ needs to be present in the concept class H and in the second one, there is no such constraint? What is the requirement on the true function f^t from which labels Y^t are generated?
- Can the authors explain the definition of learnability in definition 3.1? If Q(t) = O(t), then the algorithm can query in every iteration, right? How does this definition extend from the standard online learning definition?
- Is it true that this framework with timestamps can be expanded into the domain X to form an intuition? So, the function class is over (X,t) and we should think of (X,t) as our new domain and there is a fixed class of functions that are presented in H over this new domain?

**Limitations:**

Yes

---

> ### Author Rebuttal · Authors · 2024-08-07
>
> Thank you for your comments on the novelty and appeal of our work!
>
> ### Weaknesses
> **Comment 1**: Thank you for the feedback! We agree that presenting the theoretical framework and giving a tangible construction by referring to the specific examples mentioned in the introduction will make it much easier to understand.
>
> Regarding your second statement, the traditional setting of online learning can be considered as the fully-supervised scenario under a discrete data stream with respect to a concept class $H$. In this scenario, instances are fed one-by-one to a learner who observes the instance $X_t$, makes a prediction $\hat{Y_t}$, and observes the true label $Y_t$. The primary constraint is that the sequence of points and true labels,$ \{ (X_t, Y_t) \}\_{t=1}^{\infty} $
> , is realizable with respect to $H$ implying that there exists some $h^* \in H$ where $ \{ (X\_t, Y\_t)\_{t=1}^{\infty} \} = \{ (X\_t, h^*(X\_t)) \}\_{t=1}^{\infty} $.
>
> In the continuous setting, the instances and true labels form a continuous process throughout the time: $(X_t, Y_t)_{t \geq 0}$. Unlike the traditional setting where the learner receives the true label $Y_t$ after every prediction $\hat{Y}_t$, the learner in either the update-and-deploy or blind-prediction settings must query at that time to receive information about the true label. Additionally, the learner must obey a linear querying strategy so it is limited in the number of times it can query. The blind-prediction setting, a tougher learning framework than update-and-deploy, the learner makes predictions without knowledge of the current instance and only gains information about the instance and its true label through queries.
>
> We believe that an explanation like the one given above can be a helpful addition to our paper in highlighting the differences between the traditional and continuous settings.
>
> ### Questions
> **Question 1**: The update-and-deploy and blind-prediction settings are two separate, but closely related frameworks.
>
> The update in the update-and-deploy setting can be considered as "redeploying" or constructing a new predictor function for future instances from the data stream. In Algorithm $1$, the feedback from the query, the true label, updates the SOA algorithm and this new version of the SOA algorithm is then reinstated as the new predictor function. So, the feedback received from querying, the true point-label pair, can be used to update the predictor function, these two concepts are different.
>
> Regarding the requirements on the true function, the only requirement in both settings is that the sequence of instances and true labels, $(X_t, Y_t)_{t \geq 0}$, is realizable. If the realizability is with respect to a concept class $H$, then there exists a target concept $h^*$ such that $\forall t \geq 0, h^*(X_t) = Y_t$. In both settings, the true function, or also known as the target concept $h^*$, must lie in the concept class $H$.
>
> **Question 2**: Yes, so the definition of learnability as written in Definition 3.1 states that a concept class $H$ is learnable if there exists an algorithm $A$ employing a linear querying strategy such that $MB\_{\mathcal{P}(H)}(\mathcal{A}) < \infty$ where $\mathcal{P}(H)$ is simply the collection of all continuous processes that are realizable with respect to $H$. The expression $MB_{\mathcal{P}(H)}(\mathcal{A})$ represents the mistake-bound of the algorithm $A$ on $H$. Informally speaking, the mistake-bound is the worst case scenario of the integral of mistakes algorithm $A$ makes on any sequence realized by $H$. If this is finite, then we say that $H$ is learnable (doesn't make infinite error).
>
> Regarding algorithm querying, we can take a look at Algorithm $1$ which is an example of an $O(t)$ querying strategy. Given $\Delta$ as an input parameter, Algorithm $1$ samples its next query time, $t_q$, from a uniform distribution on an interval of width $\Delta$. Every iteration of Algorithm $1$ corresponds to its next query time which is on average spaced out by $\Delta/2$ units of time. An algorithm can query every iteration as long as its sequence of queries grows $O(t)$.
>
> Regarding the extension from standard online learning, the notion of querying primarily exists in the continuous setting because not every data point from a continuous process is observable. In the standard online learning setting, the learner operates on a discrete data stream modeled as a round-by-round process so every point is observable by the learner.
>
> **Question 3**: This is an important question to clarify. With the usual notation of $\mathcal{X}$ as the instance space and $\mathcal{Y}$ as the label space, the concept class $H$ represents some set of functions mapping from $\mathcal{X}$ to $\mathcal{Y}$. The timestamps $t$ simply represent when a particular instance-label $(X_t, Y_t)$ pair occurs. The realizability condition implies that there must exist a target concept $h^* \in H$ where $\forall t \geq 0, Y_t = h^*(X_t)$ so the ordering of a particular $(X_t, Y_t)$ occurring is irrelevant as long as $h^*(X_t) = Y_t$. When we extend to pattern classes $\mathcal{P}$, we can now incorporate the idea of temporal dependencies with instance-label pairs. For example, let's assume the continuous sequence $(X\_t, Y\_t)_{t \geq 0} \in \mathcal{P}$. Then, for some two timestamps $t, t' \in \mathbb{R}\_{\geq 0}, t\neq t'$, we decide to replace $(X\_t, Y\_t) = (X\_{t'}, Y\_{t'})$ and $(X\_{t'}, Y\_{t'}) = (X\_t, Y\_t)$ and keep all other timestamps identical to the original continuous sequence. It's not guaranteed that this sequence must lie in the pattern class $\mathcal{P}$ so the ordering of instance-label pairs becomes crucial.

---

### Decision · Program_Chairs · 2024-09-25

**Decision:**

Accept (poster)

**Comment:**

This paper introduces a novel framework for understanding online learning from both discrete and continuous data streams through selective querying. The authors propose two key settings: the "update-and-deploy" setting, where a learner updates a predictor based on queried data, and the "blind-prediction" setting, where predictions are made independently of the current data stream. The paper rigorously analyzes the learnability of concept and pattern classes under these settings, providing theoretical bounds and algorithms.

The paper presents a significant theoretical contribution to the field of online learning, with novel frameworks and insightful results. Despite some concerns about the clarity of presentation and the absence of empirical validation, the overall quality and originality of the work make it a valuable addition to the conference.